# Mechanisms of cilia regeneration in *Xenopus* multiciliated epithelium in vivo

Venkatramanan G Rao [ID] [1], Vignesh A Subramanianbalachandar [ID] [1], Magdalena M Magaj [ID] [1,2,3], Stefanie Redemann [ID] [1,2,3] & Saurabh S Kulkarni [ID] [1,4 ✉]

## Abstract

**Cilia regeneration is a physiological event, and while studied extensively in unicellular organisms, it remains poorly understood in vertebrates. In this study, using *Xenopus* multiciliated cells (MCCs), we demonstrate that, unlike unicellular organisms, deciliation removes the transition zone (TZ) and the ciliary axoneme. While MCCs immediately begin regenerating the axoneme, surprisingly, the TZ assembly is delayed. However, ciliary tip proteins, Sentan and Clamp, localize to regenerating cilia without delay. Using cycloheximide (CHX) to block protein synthesis, we show that the TZ protein B9d1 is not present in the cilia precursor pool and requires new transcription/translation, providing insights into the delayed repair of TZ. Moreover, MCCs in CHX treatment assemble fewer but near wild-type length cilia by gradually concentrating ciliogenesis proteins like IFTs at a few basal bodies. Using mathematical modeling, we show that cilia length, compared to cilia number, has a larger influence on the force generated by MCCs. Our results question the requirement of TZ in motile cilia assembly and provide insights into the fundamental question of how cells determine organelle size and number.**

**Keywords** Cilia; Transition Zone; Regeneration; Xenopus; IFT
**Subject Categories** Membranes & Trafficking; Musculoskeletal System

## Introduction

Cilia perform critical functions as the signaling hub (primary cilia) and force generator to create extracellular fluid flow (motile cilia) during growth, development, and homeostasis. Dysfunction of cilia can manifest into a constellation of defects like polycystic kidney disease (Klena et al, 2017; Noone et al, 2004), congenital heart disease (Klena et al, 2017), primary ciliary dyskinesia (Noone et al, 2004), retinal degeneration, etc., known as ciliopathies (Afzelius, 2004). Nevertheless, how cells assemble a cilium remains a central outstanding question in the field.

Cilium is an extracellular hairlike organelle composed of a microtubule-based cytoskeleton called the ciliary axoneme covered with a membrane (Ishikawa et al, 2021). Cilium is anchored to the cell via a basal body (modified centriole) and enclosed by a specialized ciliary membrane (Nicastro et al, 2006; Ishikawa et al, 2021). The ciliary axoneme consists of the ciliary shaft and ciliary tip. The ciliary shaft is the main body of the axoneme and is made of $9 + 0$ or $9 + 2$ microtubule doublets in sensory or motile cilia, respectively (Ishikawa et al, 2021). The structure of the ciliary tip can vary across species and ciliary types; however, it is usually made of microtubule singlets in motile cilia (Reynolds et al, 2018; Legal et al, 2023). The transition zone (TZ) is a structural junction between the basal body and the axoneme proposed to form a molecular sieve-like barrier (Kee et al, 2012; Okazaki et al, 2020). It acts like a "ciliary gate" regulating intraciliary protein traffic and ciliary protein content (van den Hoek et al, 2022; Kee et al, 2012; Li et al, 2016). To understand how cells construct such a complex structure, it is necessary to analyze the spatial and temporal order of the assembly of each component in the context of the entire cilium.

The presumption in the field is that cilia assemble bottom-up from the basal body, followed by the TZ and then the ciliary axoneme (Wu et al, 2024, 2021; Vieillard et al, 2016). Ciliogenesis is initiated once the cells exit the cycling phase. The centriole now undergoes maturation to form the basal body to template ciliary growth by recruiting ciliary vesicles to the distal end. The modified centriole at this stage may build a TZ and dock to the cell membrane or dock to the cell membrane, followed by building a TZ (Garcia-Gonzalo and Reiter, 2012; Sorokin, 1962; Reiter et al, 2012). Post TZ formation, the axoneme grows by recruitment of proteins at the base of cilia by Intraflagellar transport proteins (IFT) that ferry ciliary cargo to the tip of the cilia (Craft et al, 2015; Wingfield et al, 2017).

However, in motile cilia and MCCs, more evidence is needed because cilium assembly happens in the context of other intracellular events like basal body migration and cytoskeleton remodeling (Kulkarni et al, 2018; Gakovic et al, 2011; Molla-Herman et al, 2010), making it challenging to tease out the spatial and temporal sequence of events. Alternatively, one can study ciliogenesis by removing mature cilia and observing the cells reassemble the entire structure without the confounding signals

[1]Department of Cell Biology, University of Virginia, Charlottesville, VA 22903, USA. [2]Center for Membrane & Cell Physiology, University of Virginia, Charlottesville, VA 22903, USA. [3]Department of Molecular Physiology and Biological Physics, University of Virginia, Charlottesville, VA 22903, USA. [4]Department of Biology, University of Virginia, Charlottesville, VA 22903, USA. ✉E-mail: sk4xq@virginia.edu

from basal body maturation or cytoskeleton remodeling (Gakovic et al, 2011; Kulkarni et al, 2018; Rannestad, 1974). While it is important to remember that regeneration of cilia may not be identical to de novo assembly, cilia regeneration studies in *Chlamydomonas reinhardtii*, *Paramecium*, and *Tetrahymena*, etc., have provided significant insights into ciliogenesis, e.g., cargo transport, the presence of precursor pool, regulation of ciliary gene expression (Craft et al, 2015; Rannestad, 1974; Silflow and Lefebvre, 2001; Gogendeau et al, 2020; Machemer and Ogura, 1979). Cilia regeneration in unicellular protozoans is dynamic and can immediately begin post-deciliation. The deciliation trigger leads to immediate transcription and translation of ciliary machinery (Lefebvre et al, 1980; Lefebvre and Rosenbaum, 1986; Stolc et al, 2005). Blocking protein synthesis immediately post-deciliation in *Chlamydomonas* has revealed that the cells can use a precursor pool of proteins to build half the length of both cilia (Rosenbaum et al, 1969; Rosenbaum and Child, 1967; Rannestad, 1974). Partial deciliation of one of the cilia leads to the regeneration of lost cilia at the expense of the existing cilia, where it undergoes resorption till it reaches the same length as the newly built cilia. Once both attain the same length, both cilia grow to pre-deciliation length. Interestingly, blocking protein synthesis does not change the pattern of regeneration except that both cilia cannot attain pre-deciliation lengths due to the lack of new material (Rosenbaum and Child, 1967; Rosenbaum et al, 1969).

The deciliation-regeneration approach offers clear advantages to understanding ciliogenesis. However, in vertebrates, this has been hindered by a few challenges. First, ciliated cells, including cells with primary cilia or multiple cilia (MCCs), are often present in tissues that are not accessible for inducing controlled deciliation and offer technical challenges to studying regeneration by imaging. The use of cell culture could circumvent this challenge. However, a detailed study in which primary cilia or MCCs have been deciliated and regenerated is absent in the literature. These obstacles can be overcome by studying MCCs of the *Xenopus* embryonic epidermis, which are external and amenable to controlled deciliation, and studying cilia regeneration by imaging (Werner and Mitchell, 2013; Kulkarni et al, 2021). Previous studies have demonstrated that *Xenopus* multiciliated epithelium can be deciliated using the $Ca^{2+}$ shock in an in vivo setting, and these cilia regenerate within hours (Werner and Mitchell, 2013; Hibbard et al, 2021; Sim et al, 2020). Using this in vivo vertebrate system of deciliation and regeneration of cilia, our study is the first to explore the fundamental questions of spatial and temporal dynamics of motile cilia assembly.

## Results

### Cilia regeneration on *Xenopus* mucociliary epithelium

The MCCs of the *Xenopus* embryonic epidermis can be consistently deciliated within 10–15 s of exposure to a deciliation buffer, as confirmed by the loss of acetylated (Ac.) α-tubulin signal and scanning electron microscopy (SEM) (Fig. 1A,B). The recovery of Ac. α-tubulin signal was observed within an hour, indicating the initiation of cilia regeneration, with cilia reaching wild-type length by 6 h (Fig. 1C). Compared to biciliate *Chlamydomonas*, where both cilia can attain full length in 90 min, the regeneration of cilia in MCCs of *Xenopus* takes longer, raising questions about the

underlying mechanisms. One possibility is that the delay in cilia regeneration in MCCs may be due to the requirement to reassemble hundreds of cilia instead of just two in *Chlamydomonas*. Alternatively, the entire MCC may be replaced by a stem cell population rather than regenerating cilia.

The mucociliary epithelium of the *Xenopus* embryonic epidermis is similar in structure and function to the mammalian airway. Damage to cilia or MCCs in the mammalian airway is suggested to be repaired by replacing the damaged MCCs using basal stem cells (Tadokoro et al, 2014; Parekh et al, 2020). Regeneration of cilia per se in mammalian airway MCCs has not yet been reported in the literature to the best of our knowledge. Therefore, we considered two hypotheses: First, *Xenopus* MCCs regenerate cilia, or second, MCCs differentiate from the basal stem cell layer to replace damaged MCCs.

We performed live imaging of cilia regeneration to distinguish between the two hypotheses. We used *Xenopus* stem cell explants (animal caps), which auto-differentiate into an embryonic multiciliated epidermis. We harvested animal caps from *Xenopus* embryos injected with membrane RFP (to label cell and ciliary membrane) and grew them on fibronectin-coated slides. We deciliated the animal caps using the same protocol as the embryos and immediately began imaging MCCs using confocal microscopy. All MCCs showed a complete absence of cilia, indicating successful deciliation. Approximately 15 min post-deciliation, we could observe membrane RFP signal corresponding to regenerating cilia in MCCs. MCC specification, differentiation, radial intercalation, basal body amplification, migration, docking, and cilia formation from basal stem cells in *Xenopus* epithelium can take several hours (Personal observation). Thus, our results indicate that *Xenopus* multiciliated epithelium regenerates cilia in the same MCCs and does not undergo stem cell-based renewal of damaged MCCs. (Fig. EV1A; Movie EV1).

### Deciliation affects the F-actin network but does not affect basal body number and polarity

Next, we investigated whether the deciliation affects only the ciliary axoneme or cilia-associated structures, such as the apical F-actin network, basal body number, and polarity. Our results show that deciliation significantly affected the apical F-actin network (calculated as cortical and medial actin) (Fig. EV1B). We used Chibby or Centrin to label basal bodies and Clamp to label rootlets to assess the effect of deciliation on basal body number and polarity (Kulkarni et al, 2021). Our results show that deciliation does not affect basal body number and polarity. (Fig. EV1C,D).

### The transition zone (TZ) is removed by deciliation

Next, we determined the location where the deciliation treatment severed cilia. Unicellular models such as *Chlamydomonas, Paramecium*, and *Tetrahymena* lose cilia distal to the TZ and below the central pair (CP) microtubules (Sanders and Salisbury, 1994). Therefore, we hypothesized that *Xenopus* MCCs adopt a similar mechanism and would lose the ciliary axonemes distal to the TZ. To test the hypothesis, we labeled the TZ with an antibody to B9d1, a bona fide TZ protein (Chih et al, 2012; Tony Yang et al, 2015; Garcia-Gonzalo et al, 2011; Li et al, 2016; Dowdle et al, 2011). To our surprise, B9d1 was lost from the deciliated MCCs (Fig. 2A, 0 h

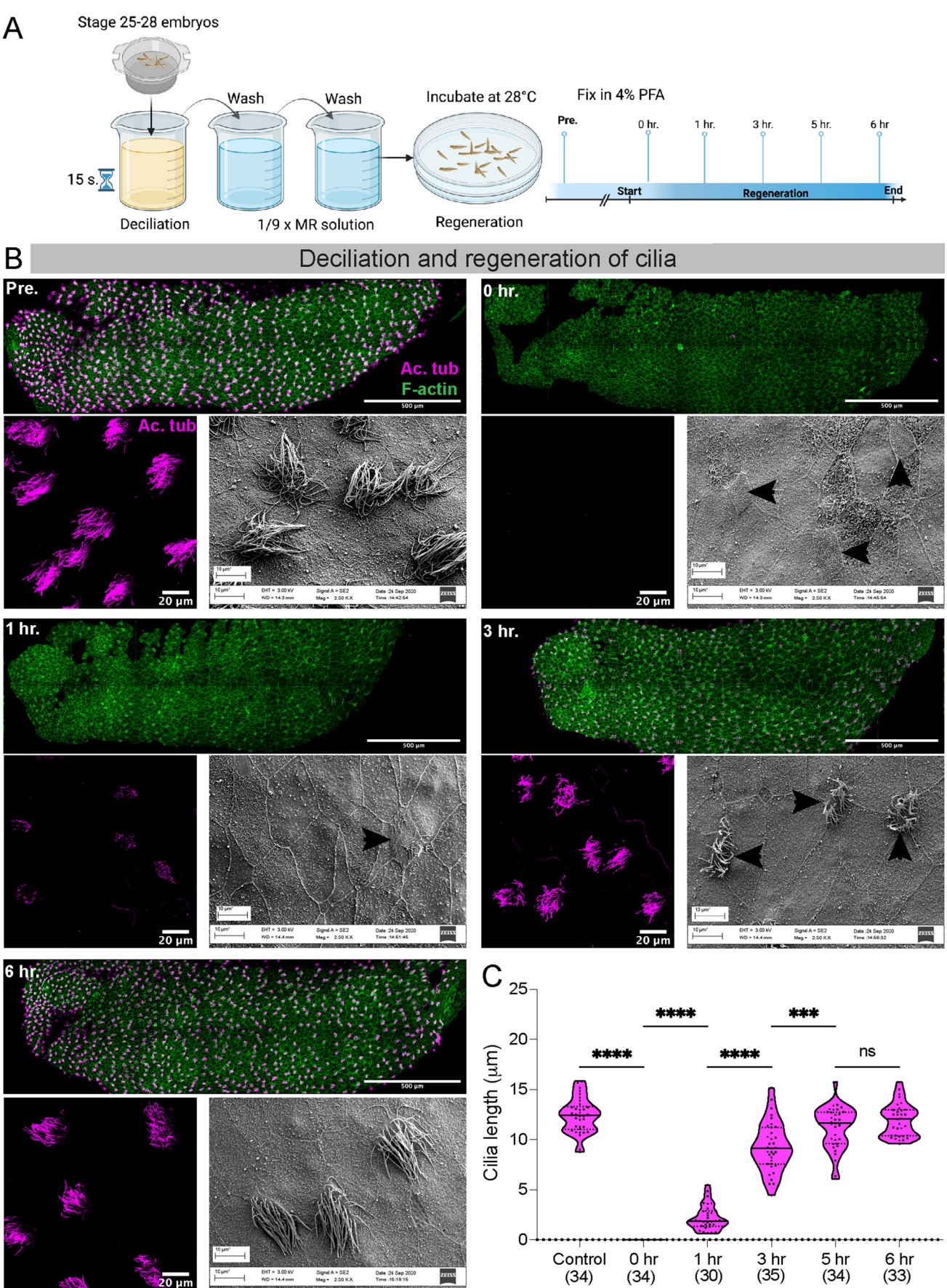

Figure 1.   **Deciliation and regeneration of cilia from the mucociliary epithelium of *Xenopus tropicalis*.**

(**A**) Cartoon depiction of the deciliation and regeneration protocol of *X. tropicalis* embryos. (**B**) Pre-deciliated embryos (Pre) and embryos during regeneration (0 h, 1 h, 3 h, 6 h) stained for cilia (Ac. α-tubulin in magenta) and F-actin (phalloidin in green). For each timepoint, the top panel shows the entire embryo, the lower left panel is a zoomed image of the multiciliated epithelium, and the lower right panel is an SEM image of the embryonic epithelium. Black arrows in the SEM image point to MCCs. (**C**) Cilia length pre- and during regeneration. The values in parenthesis indicate the number of cilia measured from three trials using 10 embryos. ****$P < 0.0001$, ***$P < 0.001$, values on top of the comparison bar denote $P$ value. ANOVA analysis was followed by Tukey's multiple comparison test. Source data are available online for this figure.

timepoint), suggesting that TZ was removed along with the ciliary axonemes. We did Transmission Electron Microscopy (TEM) to visualize the TZ and validate this unexpected result. TZ can be seen as an electron-dense H-shaped structure above the basal bodies (Figs. 2B and EV2E, Pre) (Ringo, 1967; Portman et al, 1987; Awata et al, 2014; Greenan et al, 2020; van den Hoek et al, 2022; Diener et al, 2015). One of the limitations of the TEM approach is that the sections are very thin (~75 nm), and we could miss the H-shaped structure when looking at a single section. To address this concern, we performed electron tomography of control (pre-deciliated) embryos and demonstrated the same H-shape structure is present in the ciliary axonemes (Movies EV2 and 3). we could see the basal bodies; however, the H-shaped structure corresponding to the TZ was lost entirely (Figs. 2B and EV2F, 0 min timepoint). Taken together, these data demonstrate that, unlike unicellular organisms, deciliation in *Xenopus* MCCs removes the TZ with the ciliary axoneme. While the TZ is lost with deciliation, the actual excision may happen at two distinct locations, distal to the basal body (between the basal body and TZ) and distal to the TZ (between TZ and CP), separately releasing the TZ and the axoneme into the medium (Craige et al, 2010a; Parker et al, 2010; Diener et al, 2015). Alternatively, deciliation may happen only distal to the basal bodies, in which case the TZ stays with the ciliary axoneme (Craige et al, 2010a; Awata et al, 2014; van den Hoek et al, 2022; Diener et al, 2015; Gogendeau et al, 2020). A previous proteomic study where authors deciliated *Xenopus* embryonic multiciliated epithelium found TZ proteins in their ciliary axoneme preparation, suggesting that the TZ is removed with the ciliary axonemes in *Xenopus* MCCs, supporting the latter hypothesis (Sim et al, 2020). We examined whether the B9d1 signal was associated with cilia post-deciliation to test the hypothesis. To that end, we performed an in-situ deciliation of embryos using deciliation buffer on poly-L-lysine-coated coverslips. Once removed, cilia stick to the coverslip, stained for B9d1 and Ac. α-tubulin (Fig. EV2A,B). Surprisingly, we did not see any B9d1 signal with cilia (Fig. EV2B, chemical deciliation). Notably, B9d1 is a soluble protein closely associated with the membrane-associated MKS protein complex in the TZ (Okazaki et al, 2020; Chih et al, 2012; Garcia-gonzalo et al, 2012). We examined ciliary axonemes with EM to examine if the detergent in the deciliation buffer is stripping the ciliary membrane and leading to the loss of the B9d1 signal. As expected, we could not observe any ciliary membrane (Fig. EV2D) supporting our hypothesis that loss of B9d1 is likely the result of loss of ciliary membrane. Therefore, to understand if the TZ remains associated with the axoneme after deciliation, we adopted an alternative approach of mechanical deciliation (Kiesel et al, 2020) (Fig. EV2A). The advantage of this approach is that we did not use any detergent. We observed that cilia removed from the embryo consistently showed a B9d1 signal (Fig. EV2C). Thus, we demonstrate that the site of deciliation in *Xenopus* cilia is distal

to the basal body, and the TZ remains attached to the ciliary axoneme.

## TZ is dispensable for the initiation of cilia assembly during regeneration

Removal of the TZ during deciliation gave us a unique opportunity not afforded by any other vertebrate or invertebrate model to study the temporal relationship between TZ and axoneme assembly. The TZ is thought to form a molecular sieve-like barrier and serve as a "ciliary gate," regulating protein traffic into and out of the cilium (Kee et al, 2012; Okazaki et al, 2020). Mutations in or knock-down/out of TZ proteins impair cilia assembly and function (Wang et al, 2022; Garcia-Gonzalo et al, 2011; Li et al, 2016; Craige et al, 2010b; Awata et al, 2014; Chih et al, 2012). Interestingly, the *Paramecium* also displays constant cilia shedding when TZ proteins are depleted (Gogendeau et al, 2020). Further, recent studies have shown that Intraflagellar transport (IFT) trains assemble at the TZ, suggesting that TZ is essential for ciliary assembly and is predicted to precede ciliary axoneme assembly (Molla-Herman et al, 2010; van den Hoek et al, 2022). However, no direct evidence to support this hypothesis has been published.

Using our unique cilia regeneration model, we directly tested this hypothesis. We examined the restoration of the B9d1 signal as a marker to track the reassembly of TZ during the first three hours of cilia regeneration. As expected, both the B9d1 and Ac. α-tubulin signals were completely lost after deciliation. One hour into regeneration, we were surprised to see the recovery of the Ac. α-tubulin but no detectable B9d1 signal. The B9d1 signal began to appear after 2 h of regeneration and further increased at 3 h (Fig. 2A). We quantified the length of the recovered B9d1 signal in the MCCs. In control embryos (pre-deciliation), the length of the B9d1 signal was 1.33 μm (±0.33, SEM); this signal increased from no signal at 1 h to 0.55 μm (±0.16, SEM) at 2 h and 0.73 μm (±0.021, SEM) at 3 h after deciliation (Fig. 2C). This result was unexpected because, contrary to the previous speculation, cells appeared to initiate ciliary axoneme assembly in the absence of B9D1.

One limitation of our results is that we have used only one protein, B9d1, to track the loss and reassembly of TZ. We attempted to localize multiple TZ proteins (CEP290, MKS5, TMEM216, TMEM67, and NPHP4) using fluorescent tags or antibodies in *Xenopus*, but our attempts were unsuccessful. First, antibodies are made against human/mouse TZ proteins. Second, despite our efforts with different fixation strategies, only B9D1 appeared to localize to the TZ, whereas others did not show any signal or localized at the basal body. We observed the same issue with the overexpression of fluorescent-tagged TZ proteins, where they localize to the basal bodies. Moreover, overexpression of TZ proteins also has a significant pitfall that may affect the native

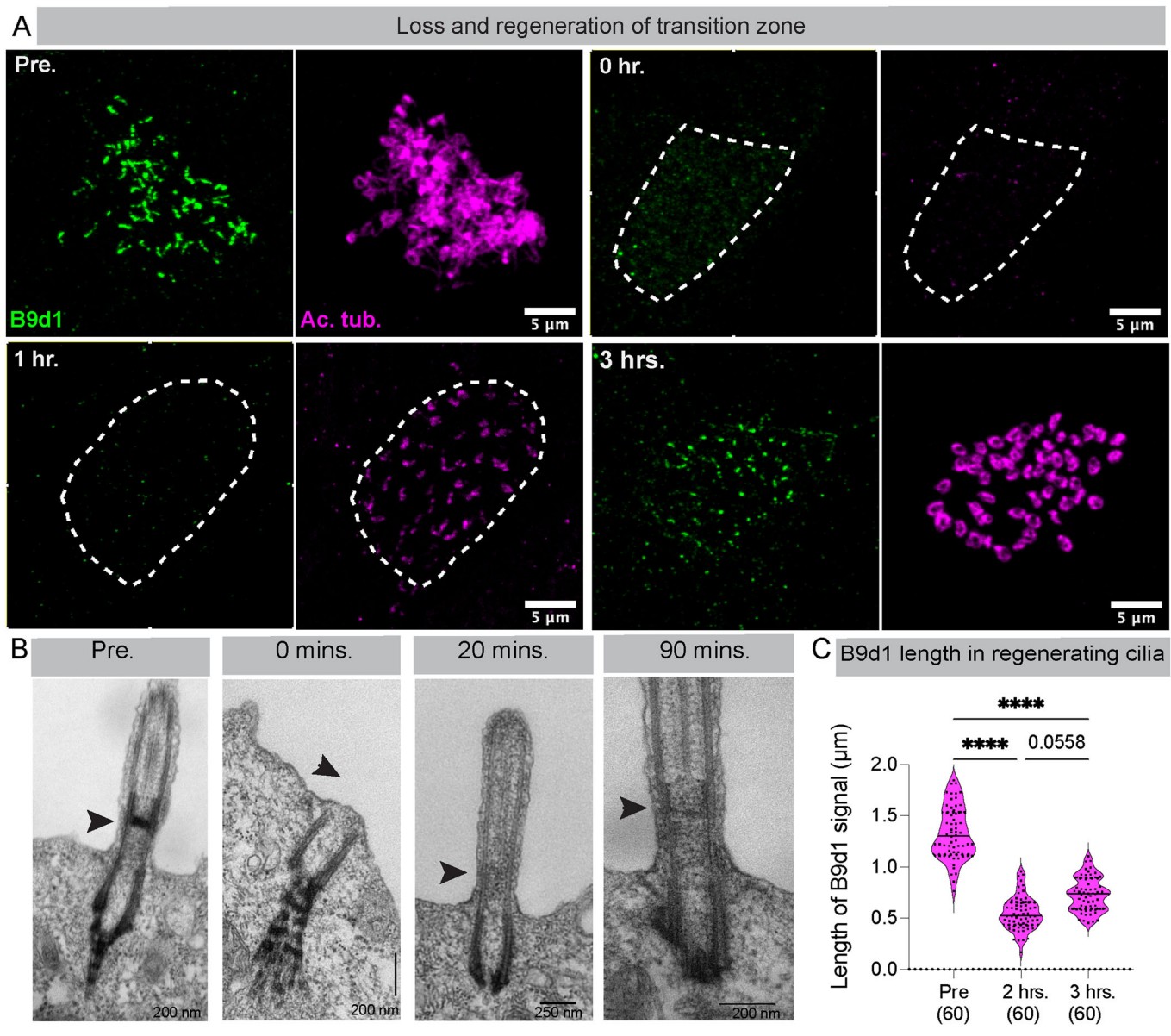

**Figure 2. Loss and regeneration of transition zone.**

(A) MCCs labeled with B9d1 antibody (TZ, green) and Ac. α-tubulin (cilia, magenta) in embryos pre- and during cilia regeneration (0 h,1 h, 3 h). MCCs are depicted by a white dashed outline. (B) Representative TEM images of cilia pre-, and during the regeneration (0 min, 20 min, and 90 min). The TZ can be seen as an "H" shaped structure between the axoneme and basal body (indicated by a black arrow). Post-deciliation samples show a complete loss of TZ. TZ is absent after 20 min of cilia regeneration. Premature electron-dense TZ structure can be seen after 90 min of cilia regeneration. (C) The length of the B9d1 signal was measured pre- and during cilia regeneration (at 2 h and 3 h) The B9d1 signal was absent at earlier time points (0 h and 1 h). The values in parenthesis indicate the number of cilia measured from three trials using 10 embryos. ****$P < 0.0001$; values on the comparison bar denote $P$ value. One-way ANOVA analysis followed by Tukey's multiple comparison test. Source data are available online for this figure.

dynamics of TZ protein localization and function during regeneration. For example, endogenous B9d1 needs to be transcribed/translated post-deciliation in *Xenopus* MCCs (see following results), and overexpression of tagged B9D1 may have altered these dynamics and the conclusions. To overcome this shortcoming, we performed electron microscopy to directly analyze the early axoneme and TZ assembly steps during regeneration. Specifically, we performed the TEM on samples taken at 0 min, 20 min, and 1.5 h post-deciliation (Figs. 2B and EV2F). At 20 min

after deciliation, a small ciliary axoneme was present; however, the H-shaped structure indicating the presence of mature TZ was not yet visible. The H-shaped structure of the TZ began to appear in the cilium 1.5 h post-deciliation. The TEM data supported the results from our B9d1 immunofluorescence experiments. To address the limitation of TEM, we performed multiple electron tomography of regenerating cilia at different time points (Pre, 0, 20 min, 1 h, 3 h, and 6 h) to examine the timing of TZ assembly (Movies EV2–14). In the two 20 min post-deciliation tomographs, we found one basal

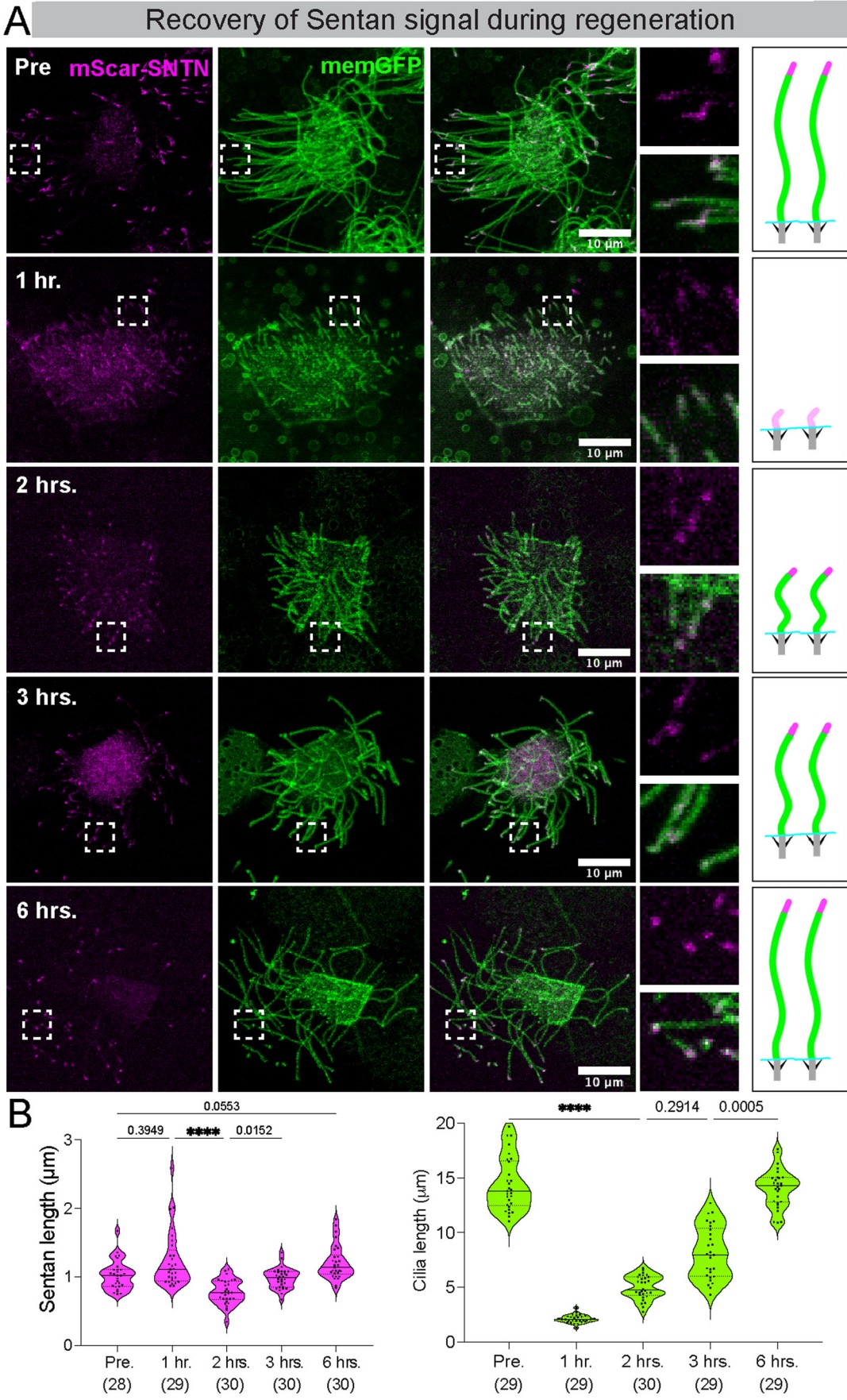

**A** Recovery of Sentan signal during regeneration

Pre — mScar-SNTN / memGFP

1 hr.

2 hrs.

3 hrs.

6 hrs.

**B**

Sentan length (μm) — 0.0553, 0.3949, ****, 0.0152

| Pre. (28) | 1 hr. (29) | 2 hrs. (30) | 3 hrs. (30) | 6 hrs. (30) |

Cilia length (μm) — ****, 0.2914, 0.0005

| Pre. (29) | 1 hr. (29) | 2 hrs. (30) | 3 hrs. (29) | 6 hrs. (29) |

**Figure 3. Ciliary tip protein Sentan is localized to the ciliary axoneme during the early stages of regeneration.**

(A) MCCs are labeled with mScarlet-Sentan decorating ciliary tips (magenta) and membrane-GFP (cilia, green) at various stages of cilia regeneration. Sentan can be seen at the tips of cilia in pre-deciliated samples. After 1 h post-deciliation, Sentan spans the entire length of ciliary axonemes (magenta transparent). At 2 h, Sentan starts accumulating at the ciliary tips. At 3 and 6 h. Sentan signal appears similar to pre-deciliated samples. (B) The Sentan signal length (left panel) and cilia length (right panel) were measured and compared among different time points. The values in parenthesis indicate the number of cilia measured from three trials using nine embryos. ****$P < 0.0001$; values on the comparison bar denote $P$ value, Kruskal–Wallis test, followed by Dunn's test. Source data are available online for this figure.

body that had not initiated axoneme assembly (Movie EV4). This was not a surprise because we see only a few ciliary axonemes at 1 h with the immunofluorescence imaging (Figs. 1B and 2A). However, the second basal body, consistent with our TEM and B9D1 data, showed a ciliary axoneme without a mature TZ structure (Movie EV5). In the seven tomographs at 1 h timepoint (Movies EV6–12), we again observed two basal bodies without ciliary axoneme. While the remaining basal bodies had ciliary axonemes with varying degrees of TZ structure, none of the TZ structures was complete, like in control axonemes. By the end of 3 and 6 h, the TZ looked indistinguishable from the controls (Movies EV13 and 14). Our results suggest that the cells can incorporate TZ (or components of TZ) once the axoneme is assembled, suggesting a dynamic nature of TZ proteins.

The structure of the ciliary axoneme is different at the proximal end, where the TZ is located, and the distal end, where the ciliary tip is located. Like the TZ, the ciliary tip comprises of a distinct protein complex of unknown function (Dentler, 1984; Portman et al, 1987; Reynolds et al, 2018; Legal et al, 2023; Sanders and Salisbury, 1994). Since the TZ showed a delayed regeneration, we asked if the ciliary tip proteins assembled after the ciliary shaft was built. To test this speculation, we used mScarlet-Sentan (Kubo et al, 2008) and RFP-Clamp (Werner et al, 2014) proteins known to localize to the tip of motile cilia. As reported earlier (Kubo et al, 2008), we observed that mScarlet-Sentan and RFP-Clamp localized to ciliary tips in control *Xenopus* MCCs (Figs. 3A and EV3A). After deciliation, ciliary Sentan and Clamp signals were lost as expected. However, when regeneration began, we noticed that small stubs of ciliary axonemes at 1 h post-deciliation were decorated with the Sentan and Clamp signals (Figs. 3A,B and EV3A,B). We measured both the Clamp and Sentan signals at different time points during regeneration. We observed that, like pre-deciliation controls, Sentan and Clamp signals became stronger at the ciliary tips as regeneration progressed. (Figs. 3B and EV3B). Our data suggests that ciliary tip proteins like Sentan and Clamp are trafficked immediately into regenerating cilia and gradually begin to concentrate at the tips.

## TZ protein B9d1 is newly synthesized during cilia regeneration

We next addressed why there was a delay in assembling TZ. The clue came from an earlier study where photobleaching of B9D1 and some other TZ proteins (in a pre-existing TZ) did not recover to pre-bleaching levels (Takao et al, 2017). This study indicated that unlike some proteins needed for axoneme maintenance, e.g., IFTs (Craft et al, 2015), the TZ proteins such as B9d1 may not be needed for cilia maintenance and thus are absent in the cytosolic pool. Consistently, we did not observe any B9d1 signal near the base of the cilia or in the cytoplasm. Therefore, we hypothesized that

post-deciliation MCCs newly synthesize TZ proteins, such as B9D1, resulting in delayed assembly of TZ.

To test this hypothesis, we treated deciliated embryos with either vehicle (DMSO) alone or Cycloheximide (CHX), a protein synthesis inhibitor. We collected embryos every hour for the first 3 h and quantified the B9d1 signal in the regenerating cilia (Fig. 4A). By 3 h, control MCCs showed a significant increase in the B9d1 signal (Fig. 4B). However, treatment with CHX completely abolished B9d1 recovery (Fig. 4A). These results suggest that B9d1 is absent in the ciliary precursor pool and requires new transcription/translation during regeneration.

## Synthesis of a new pool of ciliary proteins is required to support cilia regeneration

Our results from the previous experiments raised the question: does deciliation trigger new protein synthesis in MCCs? From the live imaging experiments, we noted that cilia regeneration begins within approximately 12–15 min after deciliation. *Chlamydomonas* also begin cilia reassembly within minutes after deciliation. Studies have shown that *Chlamydomonas* uses the precursor pool of proteins for rapid cilia reassembly; however, this precursor pool is insufficient to assemble full-length cilia (Rosenbaum and Child, 1967; Marshall and Rosenbaum, 2001). Loss of cilia triggers the synthesis of new mRNA and proteins to support subsequent cilia assembly (Marshall and Rosenbaum, 2001; Stolc et al, 2005; Lefebvre and Rosenbaum, 1986; Lefebvre et al, 1980). In contrast to *Chlamydomonas*, sea urchins contain excess proteins that can support multiple rounds (up to four times) of cilia regeneration (Auclair and Siegel, 1966). Therefore, we speculated that *Xenopus* MCCs might have a precursor pool of unassembled proteins that allow the cell to begin cilia reassembly quickly. However, it remains unknown whether *Xenopus* MCCs require new transcription and translation of some/ all proteins to complete cilia regeneration or whether the unassembled ciliary protein pool is enough to build all cilia.

To test this hypothesis, we blocked protein synthesis immediately post-deciliation using Cycloheximide (CHX); thus, the precursor pool is the only source for building new cilia. (Fig. 5A). *Chlamydomonas* has only two cilia, but in *X. tropicalis* MCCs, there are ~150 cilia. Therefore, CHX treatment could result in either the cell having a sufficient protein pool to regenerate all cilia comparable to vehicle control or a limited pool that results in MCCs choosing between the number and length of cilia.

We observed that embryos treated with CHX showed shorter cilia than controls during regeneration (Fig. 5B–D; Appendix Fig. S1A,B). Specifically, there was no significant difference in the ciliary length in the first hour between control and CHX treatment (vehicle vs. CHX, $2.75 \pm 1.04\,\mu m$ vs. $2.24 \pm 1.12\,\mu m$, mean $\pm$ SD, Fig. 5D; Appendix Fig. S1A,B). However, cilia did not grow to the same length in CHX treatment as vehicle treatment at 3 and 6 h

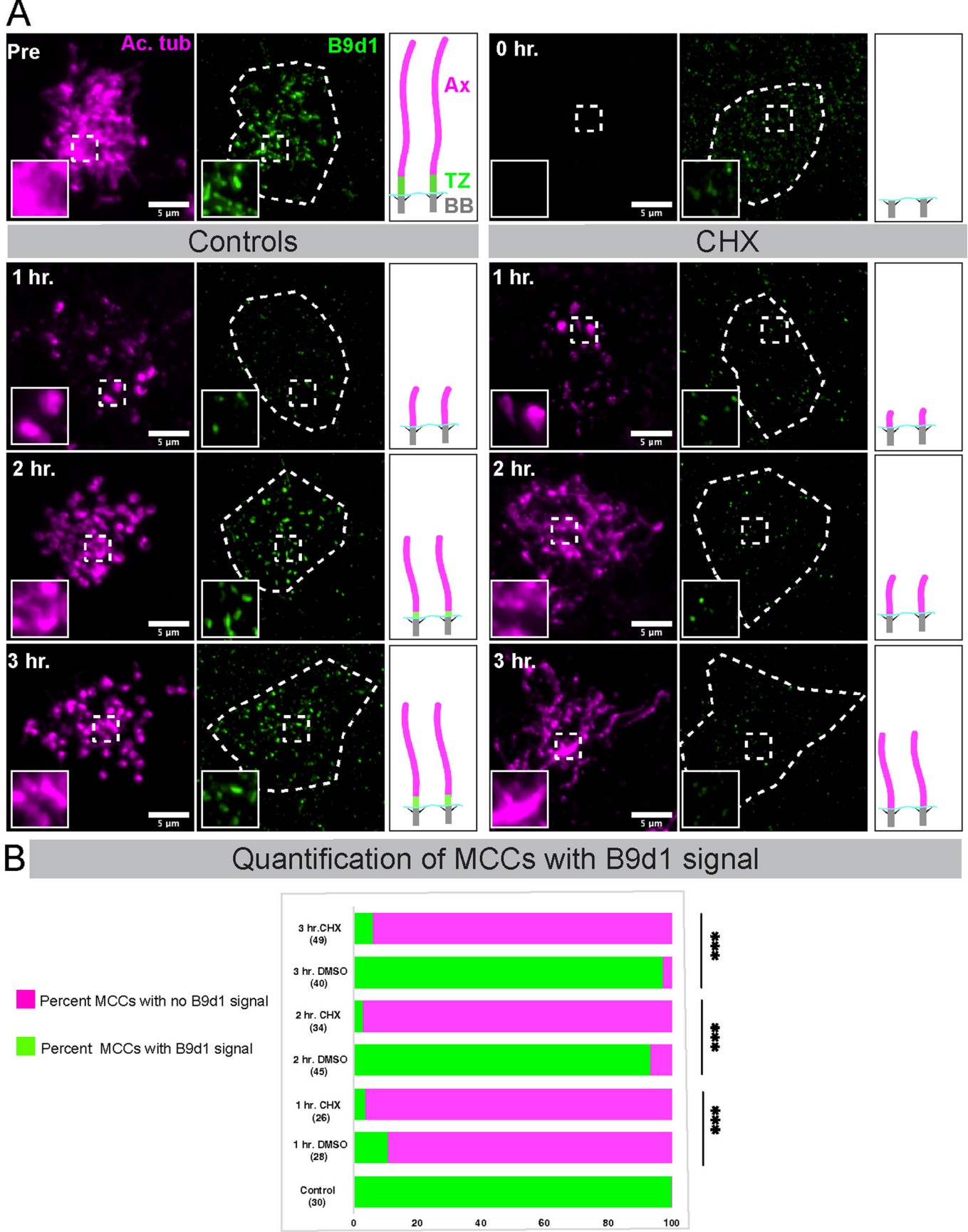

Figure 4. TZ protein B9d1 requires new protein synthesis during regeneration.

(A) Methanol-fixed Embryos treated with DMSO alone (Vehicle) or cycloheximide (CHX) in DMSO were stained for B9d1 (green, TZ) and Ac. α-Tubulin (magenta, cilia) that were collected pre-deciliation and during cilia regeneration (0 h, 1 h, 2 h, 3 h). The cell outline is marked with a white dashed line, and the dashed square box in the B9d1 channel is zoomed and depicted in the inset. While cilia can regenerate in both treatments (controls and CHX), the B9D1 signal is recovered only in controls and remains completely absent in the CHX-treated samples at 2 h and 3 h time points. Ax Axoneme, TZ transition zone, BB basal body. (B) The percent of MCCs positive for the B9d1 signal (green) and the absent B9d1 signal (magenta). Numbers in parentheses indicate the number of MCCs counted from three trials using nine embryos. ***P < 0.001, chi-square test using VassarStats to test the significance. Source data are available online for this figure.

(3 h vehicle vs. CHX, 8.53 ± 1.74 μm, vs. 5.40 ± 1.73 μm, and 6 h vehicle vs. CHX, 14.87 ± 2.38 μm vs. 8.8 ± 2.14 μm; mean ± SD; Fig. 5D and Appendix Fig. S1A,B). Thus, ciliary assembly is compromised during regeneration with CHX treatment.

Next, we measured the number of cilia per MCC during cilia regeneration. While we could count the cilia number at 1 h in both treatments (1 h vehicle 41.61 ± 12.71 vs CHX; 33.78 ± 17.27 cilia/MCC, Fig. 5C; Appendix Fig. S1C), it was challenging to count the cilia in vehicle-treated samples after 3 h and 6 h due to the long cilia that clumped/entangled during the staining process. However, the basal body number did not change with the vehicle and CHX treatment (Appendix Fig. S2A,B). Assuming all the basal bodies can assemble cilia, we can estimate 125–150 cilia/MCC in vehicle treatment (Appendix Fig. S2). In contrast to the controls, we noted that the cilia number at 1 h timepoint in CHX-treated MCCs was higher than the 3 h (1 h CHX vs. 3 h CHX; 33.78 ± 17.27 vs. 14 ± 6 cilia/MCC, Fig. 5B,E; Appendix Fig. S1C). This number was further reduced by the end of 6 h (8 ± 4 cilia/MCC; mean ± SD, Fig. 5C,E; Appendix Fig. S1C), suggesting that the CHX treatment reduced the cilia number without affecting the basal body number.

We speculated that this dramatic reduction in cilia number and length in CHX treatment could be due to the blocking of new protein synthesis and the ubiquitin-dependent proteasomal degradation of the existing ciliary protein pool in the cell (Huang et al, 2009). To test the contribution of protein degradation on cilia length and number regeneration, we blocked protein degradation using the drug MG132 (Fig. 5A) (Tasca et al, 2021). Treating the cells with MG132 with CHX did not significantly increase the number of cilia at any timepoint (Fig. 5C,E; Appendix Fig. S1C). However, CHX + MG132 led to a significant increase in cilia length compared to CHX treatment, by an average of 3 μm by 6 h (3 h CHX vs CHX + MG132; 5.4 ± 1.73 μm vs. 6.85 ± 1.53 μm, 6 h CHX vs CHX + MG132; 8.8 ± 2.14 μm vs. 11.93 ± 2.53 μm; mean ± SD; Fig. 5C,D; Appendix Fig. S1A,B).

Overall, these experiments reveal an interesting observation: with a limited protein pool, MCCs prefer longer cilia over more cilia. We speculated that MCCs optimize the length and number of cilia to generate optimal beating force. Therefore, the reason for MCCs to prefer length over number was unclear. A longer cilium can exert more beating force over a shorter cilium up to a certain length. However, increasing length beyond a threshold can increase the hydrodynamic drag (Bottier et al, 2019; Khona et al, 2013). However, in MCCs, this calculation is complex because both cilia length and number play important roles in generating the optimal force necessary for mechanical coupling with surrounding MCCs (Boselli et al, 2021). To understand this relationship, we used mathematical modeling to generate a force matrix for different numbers and lengths of cilia per cell (Dataset EV1). Our modeling

data shows that the force exerted by the beating cilia increases more for a unit increase in length than the number. For example, one 12-μm-long cilium will exert higher force than 12 cilia of 1-μm length (Dataset EV1). When applied to the number and lengths in our data, we show that the theoretical force generated by vehicle-treated MCCs increased during regeneration. Further, as expected, it was higher than the CHX and CHX + MG132 treated MCCs (Fig. 6A,B; Dataset EV1). Within the CHX-treated samples (CHX and CHX + MG132), there was an increase in force generation from 1 to 3 h, and 3 to 6 h despite a gradual decrease in the number of cilia. (Appendix Fig. S1C; Fig. 6A,B). Interestingly, the reduction in cilia number is correlated with an increase in cilia length, supporting the theoretical calculations that force generation has a quadratic relationship with cilia length compared to a linear relationship with cilia number. However, one could argue that MCCs have more cilia building blocks as a precursor pool. As a result, over 6 h of regeneration, there was an increase in total cilia quantity, which led to increased force generation (cilia quantity = avg. number of cilia X avg. cilia length). Therefore, we measured the total cilia quantity over time in CHX and CHX + MG132 treatments. Our data shows that the total cilia quantity in CHX treatment remained unchanged (75.66 μm at 1 h, 77.16 μm at 3 h, and 72.78 μm at 6 h), suggesting that the ciliary proteins were likely redistributed from shortened cilia to support fewer but longer cilia. This phenomenon was also observed in *Chlamydomonas* when one of the two flagella is removed in the presence of CHX, and the intact flagellum is shortened to support the growth of the amputated flagellum (Rosenbaum et al, 1969; Craft et al, 2015). On the other hand, there was a slight increase in total cilia quantity over time in CHX + MG132 treatment (89.56 μm at 1 h, 108.41 μm at 3 h., and 122.64 μm at 6 h). However, this increase was minimal (21.04% from 1 h to 3 h and 13.12% from 3 to 6 h) compared to the calculated force generation (214.7% from 1 to 3 h and 163.06% from 3 to 6 h), which was more proportionate to the change in cilia length (241.7% from 1 to 3 h and 174.41% from 3 to 6 h). Therefore, we propose that cilia length is a major determinant of force generation. While our model is based on the previously published literature (Bottier et al, 2019; Khona et al, 2013), it is still a simplified model and does not consider other factors, such as the coordination of beating cilia, the mucus viscosity, and the spacing/distribution of ciliary arrays on the cell, which could affect the effective force transmission.

Our model would also benefit from a direct test to measure bead movement as a readout of force generation by these MCCs. However, there are technical challenges involved in accurately measuring the bead movement over the surface of single MCCs with different cilia lengths and numbers. Even if we overcome this challenge, we do not understand when the motility machinery is fully incorporated into a regenerating cilium. Nonetheless,

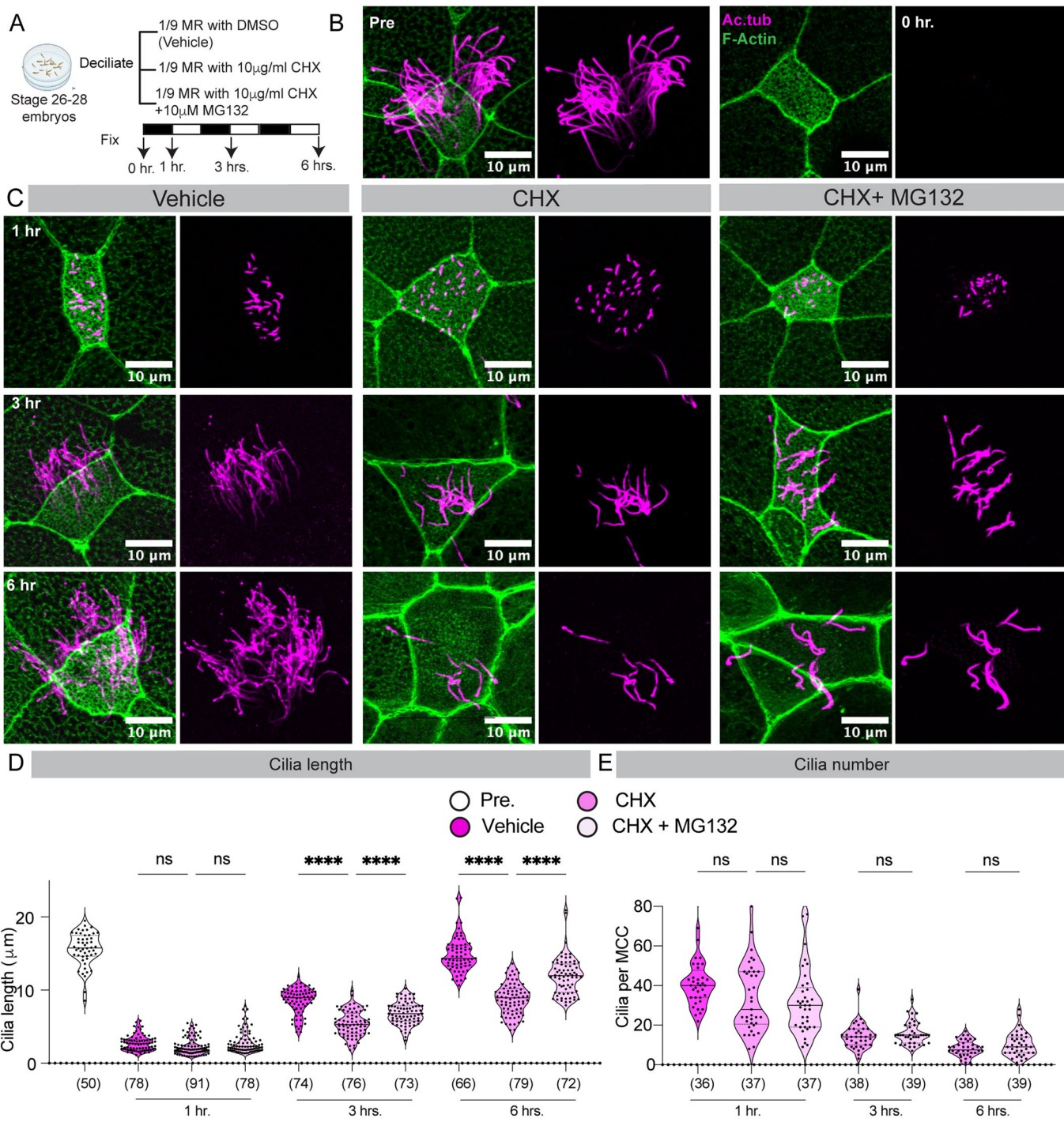

**Figure 5. Complete cilia regeneration requires new protein synthesis.**

(A) Deciliated embryos were treated with DMSO (Vehicle), cycloheximide (CHX) in DMSO, or CHX with MG132 in DMSO and collected during different time points of cilia regeneration (0 h, 1 h, 3 h, 6 h). (B) Pre-deciliated and 0 h deciliated MCCs stained for with Ac. α-tubulin antibody (cilia, magenta) and Phalloidin (F-actin, green). (C) MCCs treated with vehicle, CHX, and CHX + MG132 at 1 h, 3 h, and 6 h. (D) Graph showing the cilia lengths at different times pre-deciliation and during cilia regeneration in Vehicle, CHX, and CHX + MG132. The number in parenthesis indicates the number of cilia counted from three trials using 9–10 embryos. ****P < 0.0001, values on the comparison bar denote P value, Kruskal–Wallis test, followed by Dunn's multiple comparison test. (E) Graph showing the effect of CHX and CHX with MG132 treatments on cilia number per MCC during regeneration. The number in parenthesis indicates the number of MCCs counted from three trials using 9–10 embryos. ns not significant, Kruskal–Wallis test, followed by Dunn's multiple comparison test. Source data are available online for this figure.

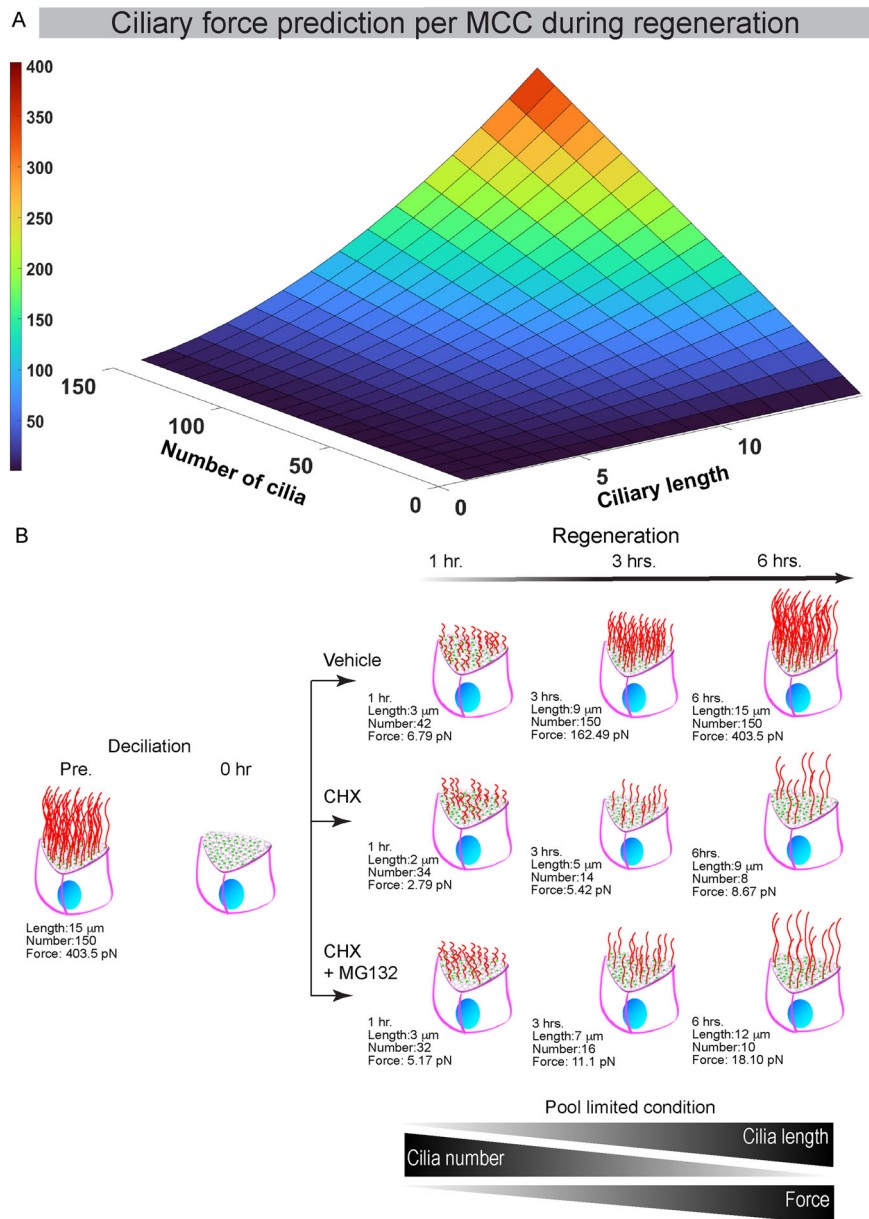

**Figure 6. Mathematical model describing the contribution of cilia length and number to force generation in MCCs.**

(A) The visualized 3-D plot shows a linear relationship between the number of cilia and the force generated by MCC, while a quadratic relationship exists between the length of cilia and the force generated by MCC. The individual force values for different cilia lengths and numbers are given in Dataset EV1. (B) The cartoon depiction of the data shows that a decrease in cilia number and an increase in cilia length was correlated with increased force generation (calculated from the model) over time in both CHX and CHX + MG132 treatments.

mathematical modeling coupled with the experimental data on cilia length and number sheds light on a fundamental cell biology question of how cells balance the number and size of organelles to optimize functional output.

## MCCs redistribute proteins among basal bodies to regenerate fewer but longer cilia

Given that the total cilia quantity in CHX treatment remains unchanged from 1 to 6 h, and MCCs cannot synthesize new

proteins, we hypothesized that existing proteins essential for ciliogenesis accumulate at a few basal bodies to support fewer longer cilia. A similar phenomenon was observed previously in *Chlamydomonas*; when *Chlamydomonas* loses one of its two flagella, it resorbs the longer flagellum to redistribute the material to the regenerating flagella to support its assembly until both flagella reach the same length (Rosenbaum and Child, 1967; Craft et al, 2015). However, unlike *Chlamydomonas* cells, *Xenopus* MCCs have hundreds of cilia, and basal bodies are not physically connected, raising the possibility of distinct mechanisms.

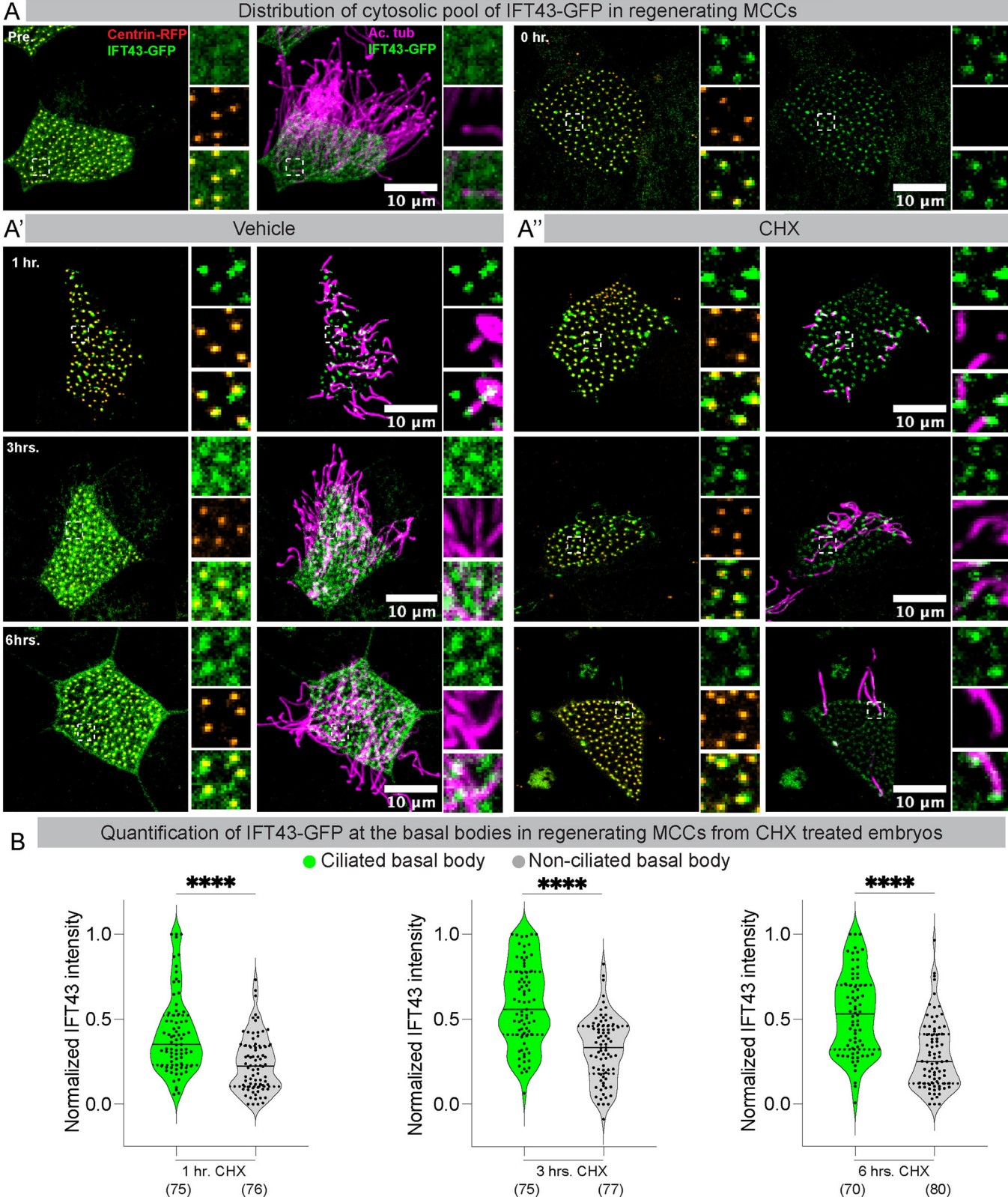

**A** Distribution of cytosolic pool of IFT43-GFP in regenerating MCCs

Pre. Centrin-RFP / IFT43-GFP | Ac. tub / IFT43-GFP | 0 hr.

**A'** Vehicle

1 hr.

3hrs.

6hrs.

**A''** CHX

1 hr.

3hrs.

6hrs.

**B** Quantification of IFT43-GFP at the basal bodies in regenerating MCCs from CHX treated embryos

● Ciliated basal body ● Non-ciliated basal body

1 hr. CHX
(75)    (76)

3 hrs. CHX
(75)    (77)

6 hrs. CHX
(70)    (80)

Normalized IFT43 intensity

****

**Figure 7. Distribution of ciliary precursor pool (IFT43-GFP) in MCCs.**

(A) Embryos injected with IFT43-GFP (green) and Centrin-RFP (orange hot, basal bodies) were deciliated at stage 28 (0 h) and were split into two experiments (DMSO and CHX). (A′, A″) The embryos in both sets were allowed to regenerate cilia for 6 h. After fixation, the embryos were stained for cilia(magenta). Note that the control MCCs at all time points (1 h, 3 h, and 6 h) have multiple cilia regenerating, and the IFT43-GFP intensity is uniform at every basal body. In contrast, the number of regenerating cilia decreases with time in CHX-treated samples, and the IFT43-GFP is enriched at a few ciliated basal bodies. (B) The intensity of IFT43-GFP associated with ciliated (green) vs. non-ciliated (gray) basal bodies in the same MCC (in CHX-treated samples) at different time points during cilia regeneration. A total of 8–10 basal bodies per MCC (4–5 ciliated and 4–5 non-ciliated) and 5 MCCs were chosen, and the mean gray value was estimated and normalized to the maximum and the minimum values in the set. The value in parenthesis indicates the number of basal bodies analyzed (with and without IFT43-GFP) from nine embryos from three independent trials. Note the significant difference in the IFT43-GFP signal intensity at ciliated vs. non-ciliated basal bodies at all time points. ****$P < 0.0001$, Kruskal–Wallis test, followed by Dunn's multiple comparison test. Source data are available online for this figure.

IFT proteins are necessary to assemble the axoneme by transporting the axonemal proteins to the tip of the growing structure (Wingfield et al, 2017; Hibbard et al, 2021; Quidwai et al, 2021). Ciliogenesis and cilia regeneration occurs by accumulating several ciliary proteins, including IFTs at the base of cilia in *Chlamydomonas* and *Xenopus* (Hibbard et al, 2021; van den Hoek et al, 2022; Craft et al, 2015). Therefore, we examined the pool of three IFT proteins (IFT20, IFT80, and IFT43) at the base of regenerating cilia as a proxy for essential ciliogenesis proteins required for cilia regeneration in the vehicle (DMSO) and CHX-treated embryos (Figs. 7, EV4 and EV5). In pre-deciliation and 0 h deciliated MCCs, all basal bodies were decorated with IFT proteins (IFT20, IFT80, and IFT43) signal, demonstrating that IFTs were similarly distributed among all basal bodies. In vehicle-treated embryos, all basal bodies remained equally decorated with IFT signal throughout regeneration as expected (Figs. 7A, EV4A, and EV5A). In contrast, in the CHX-treated cells, we observed that IFT proteins gradually enriched at a subset of the basal bodies as regeneration progressed. Interestingly, the basal bodies that were decorated with the IFT signal were also ciliated at all time points (Figs. 7A″,B, EV4A″,B, and EV5A″,B), suggesting that in the absence of new protein synthesis, IFT proteins enrich a few basal bodies and support the assembly of long cilia. Notably, the non-ciliated basal bodies at 6 h possess some IFT-GFP signal (IFT20, IFT80, and IFT43); we speculate this could be due to multiple reasons. One, some minimal amount of IFT is always associated with basal bodies. Second, these IFT molecules are in the process of getting redistributed. Note that 6 h is sufficient to regenerate full-length cilia in vehicle-treated MCCs, but we do not know if this dynamic applies to CHX treatment. Third, this results from ectopic overexpression, where excess proteins are associated with non-ciliated basal bodies.

## Discussion

Our study investigated the mechanisms of deciliation and cilia regeneration on the Xenopus embryonic epidermal multi-ciliated cells (MCCs). This is the first detailed in vivo examination of cilia regeneration in a vertebrate model system. Our findings reveal significant differences compared to previous studies in unicellular models like *Chlamydomonas* and *Tetrahymena*, providing new insights into the spatial-temporal assembly of motile cilia in vertebrates (Johnson and Rosenbaum, 1993; Silflow and Lefebvre, 2001; Rannestad, 1974; Suprenant and Dentler, 1988; Dentler, 1984; Sanders and Salisbury, 1994; Rosenbaum et al, 1969).

One of the most remarkable observations from our study is that deciliation in *Xenopus* MCCs occurs distal to the basal bodies, removing the TZ along with the ciliary axoneme (Figs. 2, 4, and 8). This contrasts with unicellular models where deciliation typically happens distal to the TZ, leaving the TZ intact with the cell (Craige et al, 2010a; Awata et al, 2014; Diener et al, 2015; Gogendeau et al, 2020; van den Hoek et al, 2022). The removal of the TZ during deciliation was confirmed using B9D1 immunofluorescence, transmission electron microscopy (TEM), and electron tomography. Interestingly, a similar deciliation mechanism was observed in human airway cilia. MCCs obtained from nasal biopsies of control and diseased subjects, subjected to calcium-based deciliation showed partial loss of the TZ, verified with nephrocystin (NPHP) antibodies (Hellmann et al, 2024; Fliegauf et al, 2006). Similar to studies from human MCCs, our findings suggest a conserved deciliation mechanism in the mucociliary epithelium of vertebrates, providing a unique opportunity to study the spatial and temporal sequence of TZ assembly relative to the ciliary axoneme.

The complete removal of the TZ allowed us to explore whether MCCs could initiate cilia regeneration without this critical structure (Figs. 2, 4, and 8). Contrary to the prevailing notion that the TZ is essential for axoneme formation (Molla-Herman et al, 2010; van den Hoek et al, 2022), we observed that *Xenopus* MCCs begin assembling the ciliary shaft and tip before reassembling the TZ. The B9D1 signal, indicative of the TZ, reappeared only after 1 h of regeneration, whereas Ac. α-tubulin marking the ciliary axoneme was detectable much earlier. These results were supported by observing the TZ structure using TEM and multiple tomograms. This suggests that the initial stages of motile cilia regeneration can bypass the requirement of the TZ. Our findings highlight the dynamic nature of TZ proteins. The delayed incorporation of B9D1 during regeneration implies that TZ components can be assembled into pre-existing axonemes. This dynamic assembly has been reported for other TZ proteins, such as CEP290 in *Chlamydomonas*, where CEP290 is exchanged between wild-type and mutant cilia during mating (Craige et al, 2010a). Based on the previous studies and our findings, we speculate that dynamic incorporation may be a general feature of TZ proteins.

Loss of TZ has been implicated in dysmorphic ciliary structures due to loss of the barrier function leading to unregulated trafficking of ciliary/non-ciliary components (Craige et al, 2010a; Awata et al, 2014; Gogendeau et al, 2020; Wang et al, 2022). However, it must be noted that in these studies, loss of TZ was prolonged, and only terminal phenotype was observed, while in our study, the loss of TZ is temporary and dynamically assembled midway during the cilia regeneration process, probably restoring the diffusion barrier. Nonetheless, our data challenges the traditional view of the TZ as

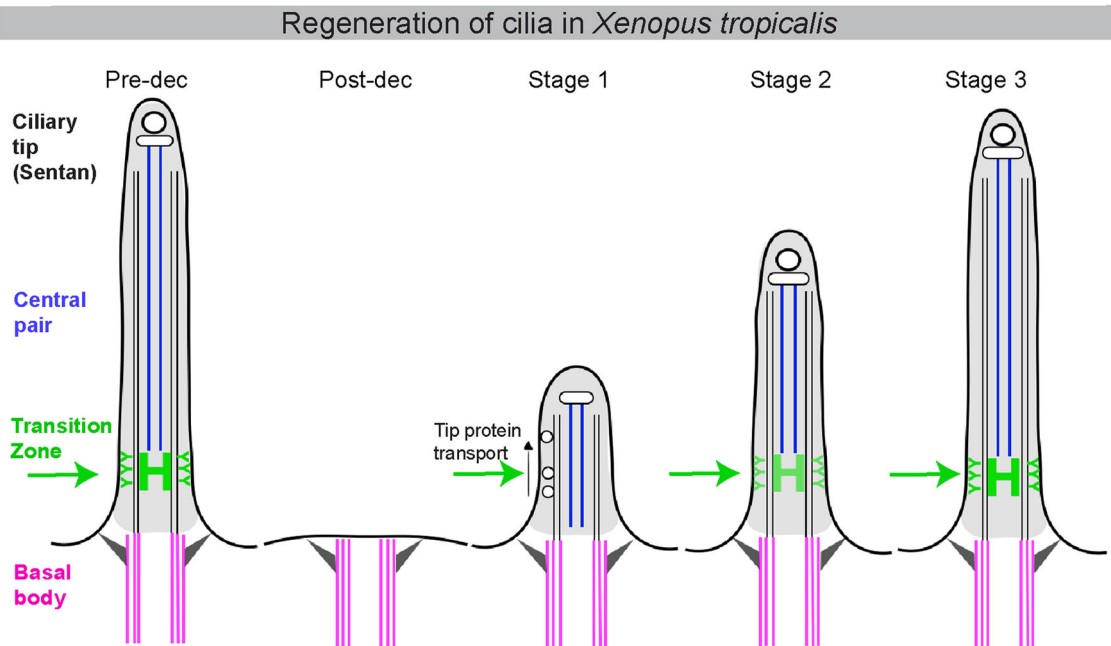

**Figure 8. Model for cilia regeneration in *Xenopus tropicalis* embryos.**

Cilia regeneration in *Xenopus tropicalis* embryonic MCCs is multi-staged and is depicted in the cartoon. Post-deciliation, the TZ is lost with cilia. During stage 1 of regeneration (0–2 h), the cilia grow without the TZ. Also, the cilia tip proteins Sentan and Clamp are immediately trafficked into the ciliary axoneme. By stage 2 (2–4 h), a partial transition zone appears, and ciliary tip proteins begin to accumulate at the distal end of the axoneme. The cilia look like pre-deciliated samples by the end of regeneration.

an essential gateway for ciliary assembly and suggests alternative pathways or mechanisms that allow axoneme formation in its absence. Our study prompts several questions critical to understanding cilia assembly and function. How do cells surpass the ciliary gating mechanism to build the axoneme? How do MCCs determine the spatial organization of the central pair microtubules without the TZ landmark? Why are the ciliary tip proteins incorporated immediately into the ciliary axoneme? When are the other components (e.g., microtubule inner proteins) critical for ciliary stability transported into the cilium?

Our experiments demonstrate that new protein synthesis is required for complete cilia regeneration. While MCCs can initiate axoneme assembly using existing protein pools, the absence of protein synthesis leads to the regeneration of fewer but longer cilia. Mathematical modeling suggests that this optimization favors cilia length over the number to maximize force generation per cell, highlighting an intriguing process that regulates organelle size and number to maintain function. While *Chlamydomonas* also redistributes proteins between its two cilia during regeneration (Rosenbaum et al, 1969; Craft et al, 2015), significant differences exist between the unicellular model and multiciliated vertebrate cells raising fascinating questions about how cells sense information and make decisions that may require quantitative data (Hendel et al, 2018; Fai et al, 2019). *Chlamydomonas* has two proximally placed and physically connected basal bodies that share protein pools, whereas *Xenopus* MCCs have hundreds of basal bodies that are not physically connected but linked via a cytoskeletal network (Kulkarni et al, 2021, 2018; Sedzinski et al, 2016). Each cilium in MCCs has its pool of proteins at the basal body, raising questions

about how proteins redistribute when resources are limited. One hypothesis is that proteins are transported by diffusion through a capture mechanism from the cytosol to the cilia (Hibbard et al, 2021, 2022). Alternatively, direct transport between basal bodies via cytoskeletal (microtubule) networks may occur (Conkar et al, 2019, 2017). Hibbard et al (2021) showed that intraflagellar transport (IFT) recruitment at basal bodies could be independent of cortical microtubules. However, this was not tested under protein-limited conditions like those induced by CHX treatment. Our results suggest that MCCs ensure at least some cilia assemble near wild-type length. We speculate that it results from a competition where cilia that have length advantage from the beginning accumulate more protein pool and become longer, and when the protein availability is limited, the shorter cilia resorb.

In conclusion, our study reveals significant new findings about the mechanisms of cilia regeneration in vertebrates, broadening our understanding of this process. The observations that cilia can initiate assembly without the TZ and that new protein synthesis is essential for complete regeneration provide valuable insights into organelle assembly and cellular decision-making processes. The insights gained from *Xenopus* MCCs offer a valuable framework for investigating the principles of organelle assembly and regeneration in complex multicellular systems. Future studies should explore the molecular mechanisms regulating the dynamic assembly of TZ proteins in the regenerating cilia. Moreover, the conservation and divergence of these mechanisms across different tissues and organisms could reveal whether similar strategies are employed in other multiciliated epithelia or disease states affecting the loss and assembly of cilia.

# Methods

### Reagents and tools table

| Reagent/resource | Reference or source | Identifier or catalog number |
|---|---|---|
| **Experimental models** | | |
| *Xenopus tropicalis* | Kulkarni et al, 2021 | RRID:NCBITaxon_8353 |
| **Recombinant DNA** | | |
| pCS2 Chibby-tagBFP2 | This study | N/A |
| pCS2 mScarlet-Sentan | This study | N/A |
| pCS2 GFP-Clamp | Kim et al, 2018 | N/A |
| pCS2 Centrin-RFP | Kim et al, 2018 | N/A |
| pCS2 MemRFP | Werner and Mitchell, 2013 | N/A |
| pCS2 IFT43-GFP | Hibbard et al, 2021 | N/A |
| pCS2 IFT20-GFP | Hibbard et al, 2021 | N/A |
| pCS2 IFT80-GFP | Hibbard et al, 2021 | N/A |
| Sentan (Accession no. NM_001321216) | Genscript | SC1200 |
| pCS2+mScarlet-C | https://www.addgene.org | RRID:Addgene_128147 |
| mTagBFP2-pBAD | https://www.addgene.org | RRID:Addgene_54572 |
| **Antibodies** | | |
| Mouse Monoclonal Anti-Acetylated α-tubulin (6-11B1) | Sigma | T6793 RRID:AB_477585 |
| Rabbit mAb Acetyl-α-Tubulin (Lys40) (D20G3) | Cell signaling technology | 5335 RRID:AB_10544694 |
| Rabbit mAb Acetyl-α-Tubulin (Lys40) (D20G3) (Alexa Fluor® 647 Conjugate) | Cell signaling technology | 81502 RRID:AB_2799975 |
| Mouse pAb Anti-B9d1 | Sigma | SAB1409114 |
| Chicken anti-mouse conjugated to Alexa fluor 488 | Invitrogen | A-21441 |
| Goat anti-mouse conjugated to Alexa fluor 488 | Invitrogen | A-11029 |
| Anti-Rabbit conjugated to Alexa fluor 647 | Jackson | 611-605-215 RRID:AB_2721876 |
| Phalloidin conjugated to Alexa fluor 488 | Invitrogen | A12379 |
| Phalloidin conjugated to Alexa fluor 647 | Invitrogen | A22287 |
| **Chemicals, enzymes, and other reagents** | | |
| MG132 | Calbiochem | 474787 |
| Cycloheximide | Sigma | 18079 |
| Poly-L lysine | Sigma | P8920 |
| NP40 alternative | Sigma | 492016 |
| Gibson assembly master mix | NEB | E2611S |
| Q5® Hot Start High-Fidelity 2X Master Mix | NEB | M0494S |

| Reagent/resource | Reference or source | Identifier or catalog number |
|---|---|---|
| **Software** | | |
| Graphpad prism | https://www.graphpad.com | RRID:SCR_002798 |
| Matlab | https://www.mathworks.com | RRID:SCR_001622 |
| Fiji | https://imagej.nih.gov/ij/index.html | RRID:SCR_002285 |
| IMOD 4.11 | https://bio3d.colorado.edu/imod/ | RRID:SCR_003297 |
| Adobe creative cloud | https://www.adobe.com/ | RRID:SCR_010279 |
| Vassar stats | http://vassarstats.net/csfit.html | RRID:SCR_010263 |
| Biorender | https://BioRender.com/z41y703 | RRID:SCR_018361 |
| **Other** | | |
| mMessage and mMachine SP6 transcription kit | Invitrogen | AM1340 |
| RNA purification kit | Zymo | R1013 |

## Animal husbandry and in vitro fertilization

Frogs (*Xenopus tropicalis*) were bred and housed in a vivarium using protocols (ACUC# 4295) approved by the University of Virginia Institutional Animal Care and Use Committee (IACUC). Embryos needed for experiments were generated using in vitro fertilization as described before (Kulkarni et al, 2021; Khokha et al, 2002). Briefly, the testes from male frogs were crushed in 1× MBS (pH—7.4) with 0.2% BSA and added to eggs obtained from the female frogs. After 3 min of incubation, freshly made 0.1× MBS (pH—7.8) was added, and the eggs were incubated for 10 more minutes till contraction of the animal pole of the eggs was visible. The jelly coat was removed using 3% cysteine in 1/9× MR solution (pH 7.8–8.0) for 6 min. The embryos were either microinjected with DNA/RNA and raised at 25 °C or 28 °C till they reached appropriate stages for fixation and staining or mucociliary organoid (animal cap) dissection. Embryos were staged as described previously (Zahn et al, 2022). The fertilized embryos for microinjection were chosen randomly.

## Animal cap/mucociliary organoid dissection

Embryos at stage 10 were used for animal cap dissection. The animal pole from the stage 10 embryos was dissected using a fine hair tool in a pool of Danilchik's for Amy (DFA) medium (Kulkarni et al, 2021). The dissected tissue was trimmed to remove the equatorial cells from the sides. This tissue was then laid on a coverslip or glass slide coated with fibronectin (Sigma Cat No. F1141) cultured in DFA solution. The tissue was allowed to grow and spread on the fibronectin-coated slides/coverslips overnight till they reached stage 28.

## DNA, RNA, and microinjections

Plasmids used in this study are listed in the reagents and tools table. To generate Chibby-BFP, the full-length Chibby gene was cloned to the N-terminus of mtagBFP2 in the pCS2+ vector (Subach et al, 2011; Kulkarni et al, 2021). Chibby and mtagBFP2 were amplified by PCR using 2x Q5 polymerase master mix and cloned in pCS2+ vector by Gibson assembly (Appendix Table S1). The human sentan gene was synthesized from Genscript. The gene was amplified using specific primers (Appendix Table S1) using Q5 polymerase and cloned into a pCS2+ mScarlet vector using Gibson assembly. RNA used in this study were generated by linearizing the plasmids and in vitro transcribed using the mMessage and mMachine SP6 transcription kit) and purified using the Zymo RNA purification kit. All synthesized RNAs were microinjected at 1-cell stage (Chibby-mtagBFP2 100 pg, mScarlet-Sentan 200 pg, EGFP-Clamp 200 pg, Centrin-RFP 100 pg, membrane RFP 100 pg, IFT43-GFP, IFT80-GFP, and IFT20 150 pg.) at one cell stage using glass needles mounted on the Pico-liter microinjection system (Warner instruments). All the microinjections were done at the 1-cell stage except for Fig. 2, where the membrane RFP was injected in 1 of the four blastomeres at the four-cell stage (30 pg).

## Chemical deciliation of embryos and mucociliary organoids and drug treatments

Embryos, after they reach stage 28, were used for deciliation experiments. A basket was designed for deciliation by cutting a 50-ml centrifuge tube. One of the open ends of the tube was heat-sealed with a muslin cloth. A six-well plate was filled with deciliation solution (75 mM $CaCl_2$, 0.02% NP40 in 1/3 MR solution). The rest of the five wells were filled with 1/9× MR. Anesthetized embryos were transferred to this container and dipped in a deciliation solution (75 mM $CaCl_2$, 0.02% NP40 alternative in 1/3 MR solution) in a six-well plate for 10–15 s. Post-deciliation, embryos with the container were dipped/washed with 1/9× MR in the rest of the wells of the six-well plate. The embryos were then transferred to a Petri dish with 1/9× MR with gentamicin and incubated at 28 °C. Embryos were collected during various stages of cilia regeneration, as mentioned in the text and figure legends. For animal caps, the glass slide/coverslip was removed from the DFA medium and dipped in the deciliation solution for 15 s. Post-deciliation, the glass side/coverslip was dipped in 1/9× MR for 30 s and transferred to DFA until the cilia were regenerated to the appropriate stage. Control or deciliated embryos were transferred to 1/9× MR solution containing either 10 µg/ml of cycloheximide (CHX) or DMSO (vehicle) for CHX experiments. MG132 was used to prevent ubiquitin-mediated protein degradation; we used 10 µM of MG132 dissolved in DMSO for treatment post-regeneration.

## Mechanical deciliation

Poly-L-lysine-coated cover glass was used for the mechanical deciliation of embryos. 25–30 embryos at NF stage 28–30 were transferred to the poly-L-lysine-coated cover glass, and any residual 1/9× MR was removed. Mechanical pressure was applied by a glass slide placed on top of the embryos for a minute. After a minute, the embryos were discarded, and the cilia stuck to the cover glass were fixed in chilled methanol for 10 min and rehydrated using 1× PBS, followed by blocking and staining with anti-Ac. α tubulin antibody and anti-B9D1 antibody.

## Immunofluorescence staining and imaging

The X. tropicalis embryos used for the study were fixed post-stage 28 with appropriate fixative agents. For most experiments, 4% paraformaldehyde (PFA) was used as a fixative, and 100% chilled methanol was used to fix the embryos for anti-B9d1 staining. After fixation, the embryos were washed three times with PBST (1× PBS with 0.2% Triton X-100) for 10 min each and then incubated in a blocking solution (3% BSA in PBST) for 1 h. Appropriate antibodies were added to the embryos, incubated for 1 h at room temperature, and rewashed three times for 10 min each with PBST. A conjugated secondary antibody was used to stain embryos for 1 h. The embryos were washed thrice with PBST, stained with phalloidin in PBST for 45 min, and washed once post-staining. The embryos were mounted and imaged. Confocal imaging was performed using the Leica DMi8 SP8 microscope with a 40x oil immersion objective (1.3 NA). Images were captured at 1× or 4× zoom and adjusted (brightness and contrast), analyzed, cropped in Fiji, and assembled in Adobe Illustrator software.

## Sample preparation for electron tomography

Xenopus stage 26–28 embryos Pre and post-deciliation were fixed in 2.5% glutaraldehyde and 2% paraformaldehyde in 0.1 M sodium cacodylate buffer pH—7.4 for 1 h, rinsed in the buffer, then post-fixed in 1% osmium tetroxide. Subsequently, embryos were rinsed in buffer and stained in 2% aqueous uranyl acetate for another hour. Embryos were rinsed and dehydrated in an ethanol series (15 min 30% ethanol, 15 min 50% ethanol, 15 min 70% ethanol, 15 min 90% ethanol, 3 × 30 min 100% ethanol). Dehydrated embryos were infiltrated with EPON Araldite (Electron Microscopy Sciences) and polymerized overnight at 60 °C. Serial semi-thick sections (200 nm) were cut using an Ultracut UCT Microtome (Leica Microsystems, Vienna, Austria). Sections were collected on piliform-coated copper slot grids and poststained with 2% uranyl acetate in 70% methanol, followed by Reynold's lead citrate.

## Data acquisition by electron tomography

Colloidal gold particles (15 nm; Sigma-Aldrich) were attached to both sides of semi-thick sections collected on copper slot grids to serve as fiducial markers for subsequent image alignment. For dual-axis electron tomography, a series of tilted views were recorded using a TECNAI F20 transmission electron microscope (FEI Company, Eindhoven, The Netherlands) operated at 200 kV. Images (4 K × 4 K) were captured every 1 h over a ± 60 range and a pixel size of 2.3 nm using a TVIPS CCD camera. We used the IMOD software package for image processing, which contains all the programs needed to calculate electron tomograms. The tilted views were aligned using the positions of the colloidal gold particles as fiducial markers. Tomograms were computed for each tilt axis using the R-weighted back-projection algorithm. Reconstructed tomograms were flattened.

## Image analysis, quantification, and statistics

All the experiments were repeated three times. All the measurements and analyses were performed on at least three embryos. Sample size, indicated by "n" values, is included in the figure legends. For length measurement (cilia length, ciliary tip length, and TZ length), at least

three cells per embryo were chosen randomly; however, it was ensured that only those cilia/signals that were distinct and not overlapping (or tangled) with neighboring ones were measured. For statistical analysis, Prism ver. 10 was used. The type of analysis, p-values, and significance are included in the figure legends. The F-actin intensity measurements were done using Fiji. The cortical actin refers to the actin network along the MCC boundary, while the medial actin refers to the F-actin network at the apical surface of the MCC. The intensity measurements were performed using a line or polygon tool in Fiji, as described in an earlier study (Kulkarni et al, 2018). For orientation calculations, images were taken after aligning the embryos horizontally from anterior on the left to posterior on the right and stained for Clamp-GFP (rootlet) and Centrin-RFP (basal body). The Clamp-GFP orientations were calculated in MATLAB R2021a using image processing and circular statistics toolboxes. The cilia orientations were drawn by hand from rootlet to basal body using the 'drawline' command. The angles were calculated with respect to the global X-axis in the cartesian coordinate system (first point starting at 0,0) using the command "atan2d", which gives the angles from 0° to +180° in a counterclockwise direction (quadrants I and II) and 0° to −180° in a clockwise direction (Quadrants IV and III). 26 and 21 multiciliated cells were analyzed for control and deciliated embryos, respectively, with an average of 112 cilia orientations ( ~ 60–200 cilia per cell) in each cell for each condition. For the IFT-GFP (IFT20, IFT80, and IFT43) fluorescence intensity calculations, we adopted an unbiased approach to measure the IFT intensities. Firstly, the three channels were separated using Fiji. The cilia and centrin channels were merged, and a circular ROI between 0.646–0.680 μm$^2$ was drawn around the ciliated and non-ciliated centrin-RFP puncta (4–5 basal bodies from each category). The mean gray value of the IFT-GFP channel was calculated, and the values were normalized from 0–1. The normalization was done using the following formula:

Normalized value = (mean gray value – min value in the dataset)/(max value in the dataset − min value in the dataset). For all the experiments, no blinding was applicable.

### Theoretical estimate of force applied by a single multiciliated cell (MCC)

Boselli et al derived a simple force model for a cilium from resistive force theory (Boselli et al, 2021) using a simplified assumption of cilium as a rigid rod rotating around the base, making an arc of varying lengths from 0 to length of the cilium (l) during each cycle. By ignoring force contribution during the recovery stroke, the effective force contribution from a cilium during a cycle is simplified as

$$f = \frac{\zeta \times l \times V_c}{12}$$

where $l$ is the length of the cilium, $V_c$ is the velocity approximated as $\sim 4 \times v \times l$ where $v$ is the beat frequency (assumed as 25 Hz and $l$ is the length of the cilium) and

$$\zeta = \frac{4\pi\mu}{\ln(\sqrt{e\varepsilon})}$$

is the transverse drag coefficient, where μ is the viscosity of the liquid (assumed as 0.001 Pa. s for water), e is the natural log

(e = 2.718281828459), $\varepsilon$ is the cilium aspect ratio (length/thickness), the cilium's thickness is assumed to be 0.2 μm.

Force contribution from the entire MCC is approximated as F = f × $N$ × $C_o$, where f is the force contribution from a single cilium, where $N$ is the total number of cilia in the cell, and $C_o$ is the coupling coefficient, which is defined as 0.55 based on Boselli et al,'s force calculations (Boselli et al, 2021). This study has shown that for far-field estimates, ~50–60% of cilia do not contribute to a cell's final average lateral force component due to entanglement and phase shifts (Boselli et al, 2021). Hence, a value of $C_o$ = 0.55 is used for the calculations, reflecting close to the actual force contribution from each cell. Using the formula described, force contribution from a single MCC was simulated for varying lengths (1–15 μm assuming all cilia in the cell have the same length) and number of cilia (1–150) and visualized as surface plots.

## Data availability

This study includes no data deposited in external repositories.

The source data of this paper are collected in the following database record: biostudies:S-SCDT-10_1038-S44319-025-00414-8.

## Peer review information

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

## Acknowledgements

The authors thank Dr. Karen Hirsch for providing access to the confocal microscope. The authors would like to express their gratitude to Dr. Bob Bloodgood and the members of the Kulkarni Lab for their invaluable feedback and thorough review of this manuscript. Appreciation is also extended to Dr. Mustafa Khokha, Yale University, for the initial discussions and intellectual guidance for the project. The authors acknowledge the Advanced Microscopy Facility supported by the University of Virginia School of Medicine, Research Resource Identifiers (RRID): SCR_018736) for SEM services (NIH SIG grant 1S10OD011966-01A1). Transmission electron micrographs were recorded at the University of Virginia Molecular Electron Microscopy Core facility (RRID: SCR_019031), partially supported by the School of Medicine and built with NIH grant G20-RR31199. Lastly, we are grateful for the NIH Pathway to Independence, K99/R00 (K99HL133606 and R00HL133606) grant, awarded to Saurabh Kulkarni and NIH-NIGMS grant (1R01GM144668-01) awarded to Stefanie Redemann.

## Author contributions

**Venkatramanan G Rao**: Conceptualization; Validation; Investigation; Visualization; Methodology; Writing—original draft; Writing—review and editing. **Vignesharavind Subramanianbalachandar**: Conceptualization; Investigation; Methodology. **Magdalena M Magaj**: Methodology. **Stefanie Redemann**: Supervision; Investigation; Visualization; Methodology. **Saurabh S Kulkarni**: Conceptualization; Resources; Supervision; Funding acquisition;

Validation; Investigation; Visualization; Methodology; Writing—original draft; Project administration; Writing—review and editing.

Source data underlying figure panels in this paper may have individual authorship assigned. Where available, figure panel/source data authorship is listed in the following database record: biostudies:S-SCDT-10_1038-S44319-025-00414-8.

## Disclosure and competing interests statement

The authors declare no competing interests.

# Expanded View Figures

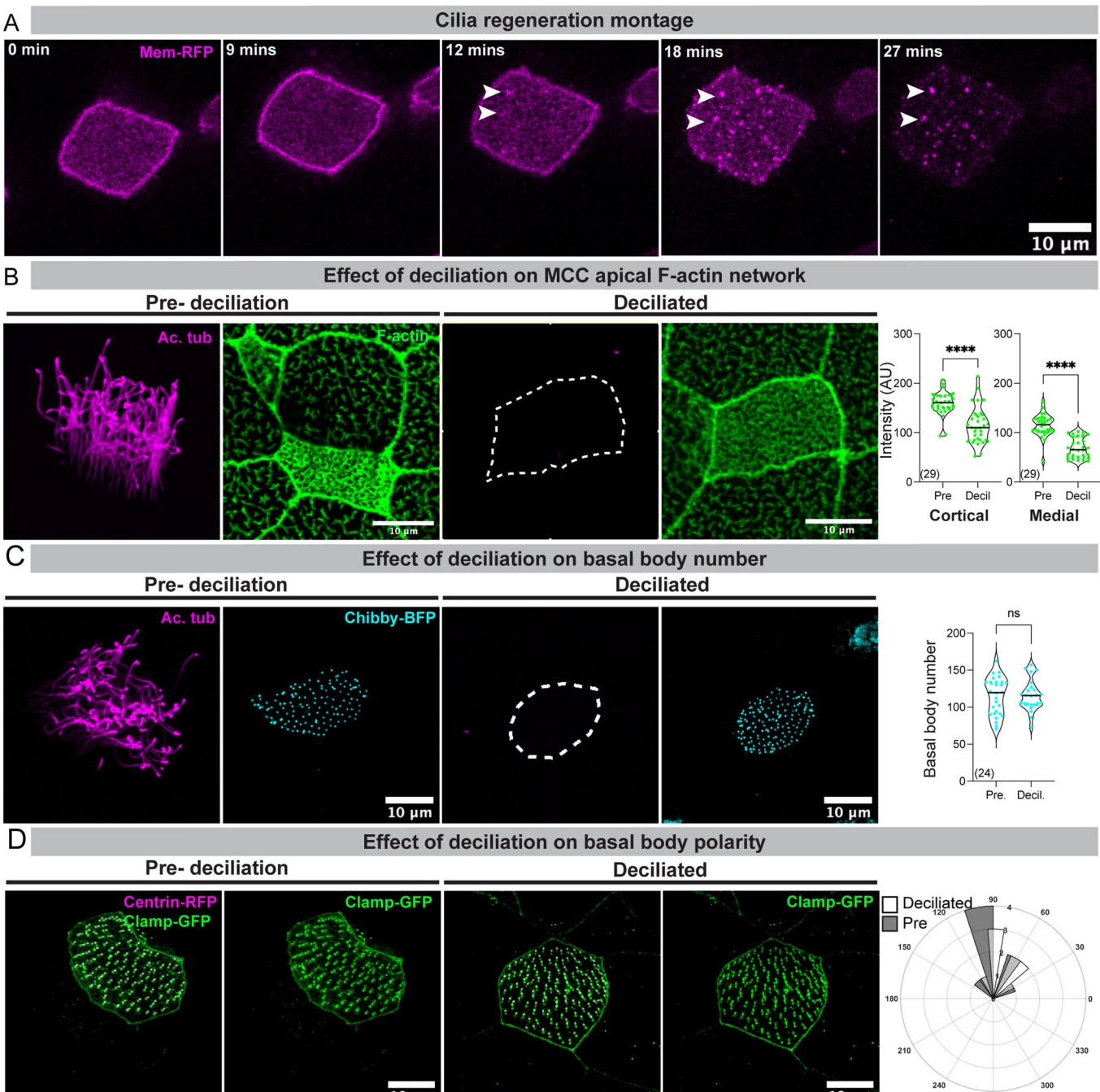

**Figure EV1. Deciliation affects apical F-actin, but the basal body number is unaffected.**

(A) Montage of regenerating cilia (mem-RFP) in the animal caps. The mem-RFP signal specks (marked by white arrows) can be seen emerging from the cell's surface by ~12 min, eventually growing into beating cilia (see Movie EV1). (B) Pre- and post-deciliation MCCs that are stained for cilia(magenta) and F-actin(green) are depicted. Cortical and medial F-actin intensity significantly differs in Pre and 0-h deciliated samples. The values in parenthesis indicate the number of MCCs measured from three trials from 9–10 embryos. ****$P < 0.0001$, Mann–Whitney test. (C) The number of basal bodies labeled with Chibby-GFP (cyan) is unaffected after deciliation. The values in parenthesis indicate the number of MCCs measured from three trials from 9–10 embryos. ns = not significant, Mann–Whitney test. (D) Deciliation does not affect basal body polarity. Basal body polarity was determined by measuring the orientation of rootlets labeled with Clamp-GFP (green) with their respective basal body labeled with Centrin-RFP (magenta) and represented in the rose plot. The directionality was measured in 27 pre-deciliated cells and 22 MCCs post-deciliation from three trials from 9–10 embryos in each category.

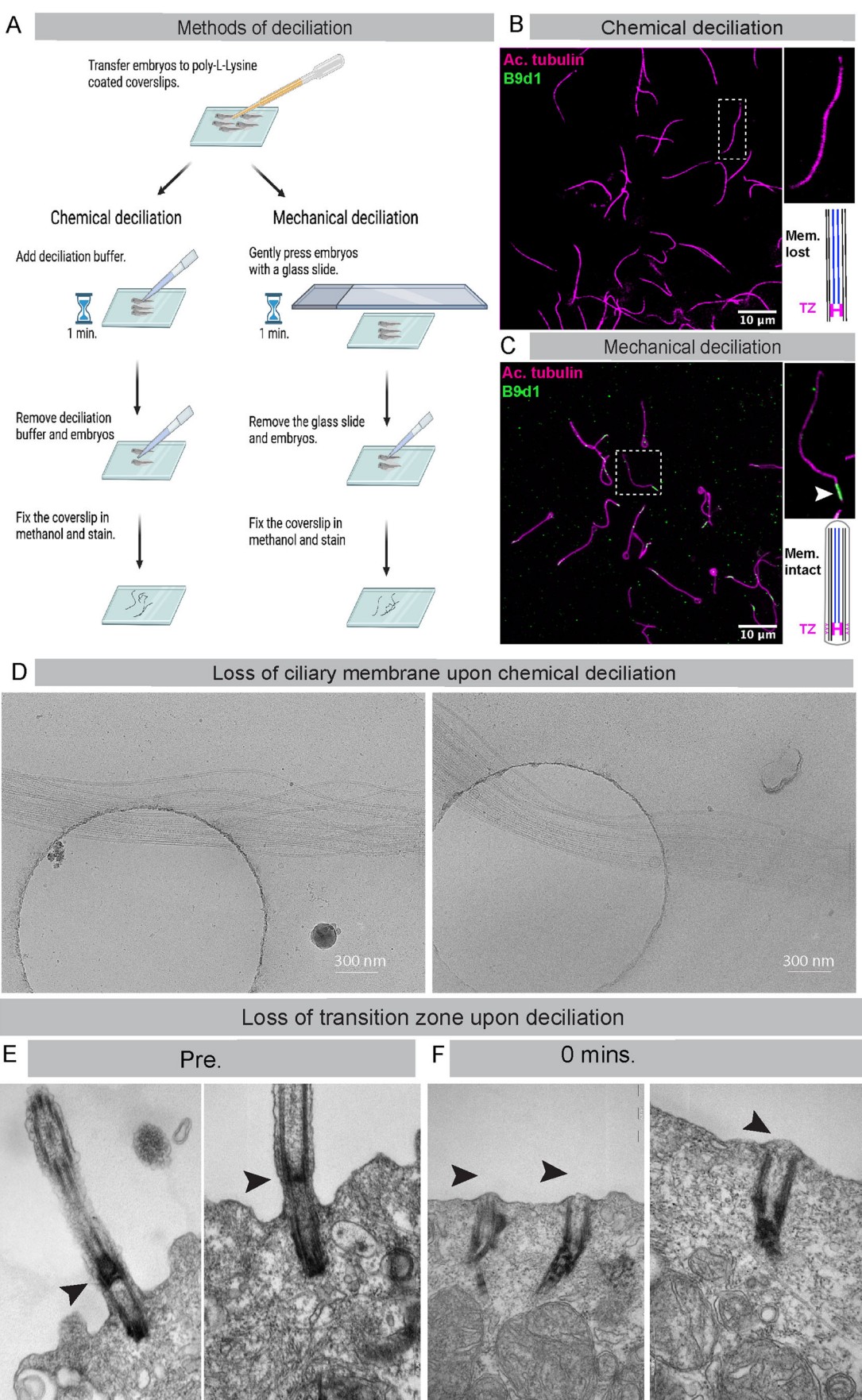

**A** Methods of deciliation

Transfer embryos to poly-L-Lysine coated coverslips.

Chemical deciliation

Add deciliation buffer.
1 min.

Remove deciliation buffer and embryos

Fix the coverslip in methanol and stain.

Mechanical deciliation

Gently press embryos with a glass slide.
1 min.

Remove the glass slide and embryos.

Fix the coverslip in methanol and stain

**B** Chemical deciliation

Ac. tubulin
B9d1

Mem. lost

TZ

10 µm

**C** Mechanical deciliation

Ac. tubulin
B9d1

Mem. intact

TZ

10 µm

**D** Loss of ciliary membrane upon chemical deciliation

300 nm

300 nm

Loss of transition zone upon deciliation

**E** Pre.

**F** 0 mins.

◄ **Figure EV2. Transition zone is removed with cilia during deciliation.**

(**A**) Schematic of the chemical and mechanical deciliation methods. (**B**) Cilia with chemical deciliation lose the B9d1 signal, possibly due to the loss of membrane with detergent in the deciliation buffer, whereas (**C**) the B9d1 signal (marked by white arrows) is maintained with mechanical deciliation. (**D**) Electron micrographs of cilia from the chemical deciliation method lack the ciliary membrane and show splaying of axonemal microtubules. (**E**) Representative TEM images of cilia pre-deciliation with intact TZ and (**F**) immediately post-deciliation (0 h), revealing the loss of TZ (arrows indicate TZ location).

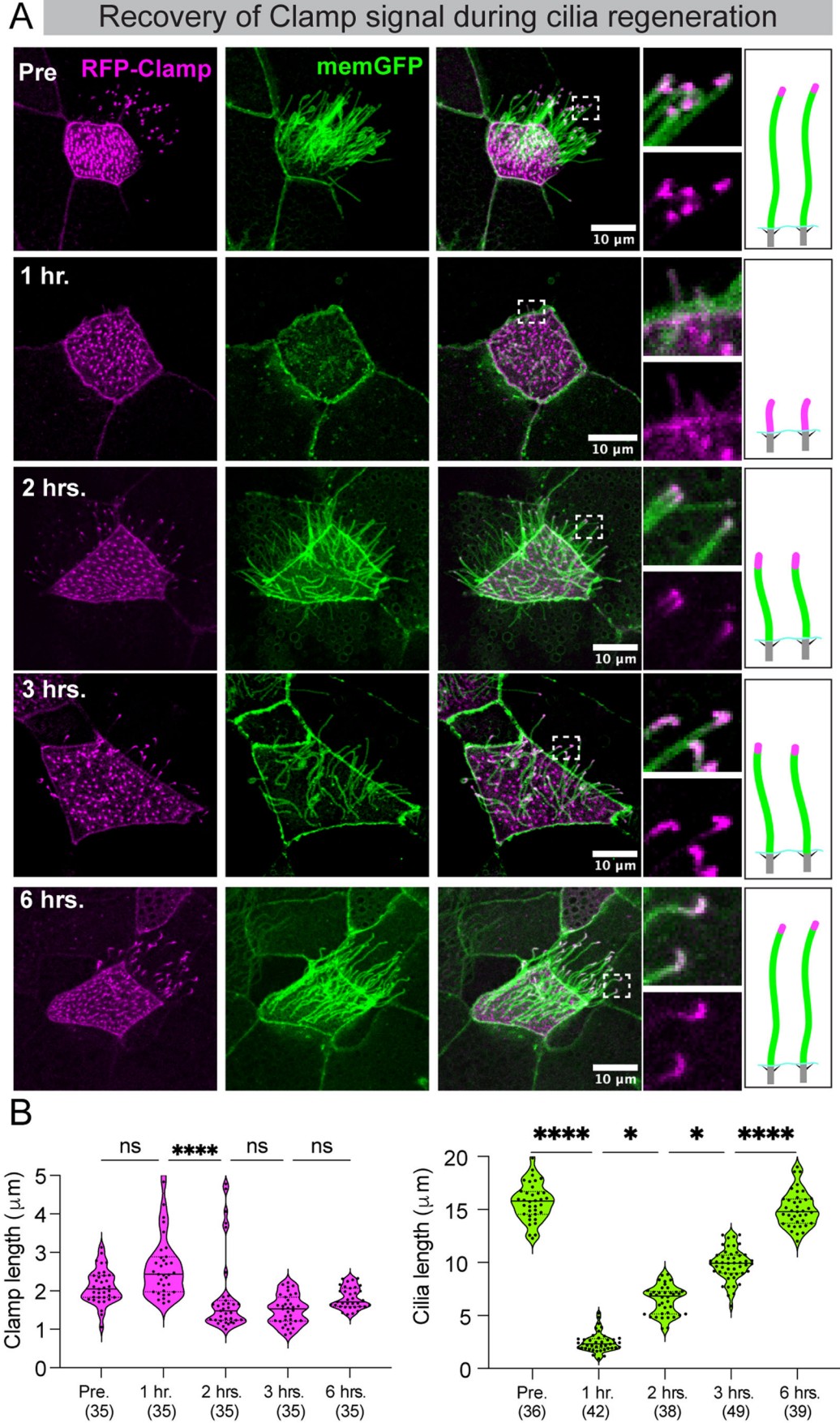

**A  Recovery of Clamp signal during cilia regeneration**

Pre  RFP-Clamp  memGFP

1 hr.

2 hrs.

3 hrs.

6 hrs.

**B**

Clamp length (μm) — ns, ****, ns, ns

Pre. (35)  1 hr. (35)  2 hrs. (35)  3 hrs. (35)  6 hrs. (35)

Cilia length (μm) — ****, *, *, ****

Pre. (36)  1 hr. (42)  2 hrs. (38)  3 hrs. (49)  6 hrs. (39)

**Figure EV3. Clamp is localized to the ciliary axoneme during the early stages of regeneration.**

(A) MCCs are labeled with RFP-Clamp (ciliary tip and base, magenta) and memGFP (cilia, green) at various stages of cilia regeneration. After 1 h post-deciliation, the Clamp signal can be seen in the ciliary axonemes (magenta transparent). At 2 h, the Clamp signal starts accumulating at the ciliary tips. At 3 and 6 h, the Clamp signal appears more like pre-deciliated samples. (B) Clamp signal length (left panel) and cilia length (right panel) were measured and compared among different time points. The values in parenthesis indicate the number of cilia measured from three trials using 9–10 embryos. ****$P < 0.0001$; values on the comparison bar denote $P$ value; ns - not significant, Kruskal–Wallis test, followed by Dunn's test.

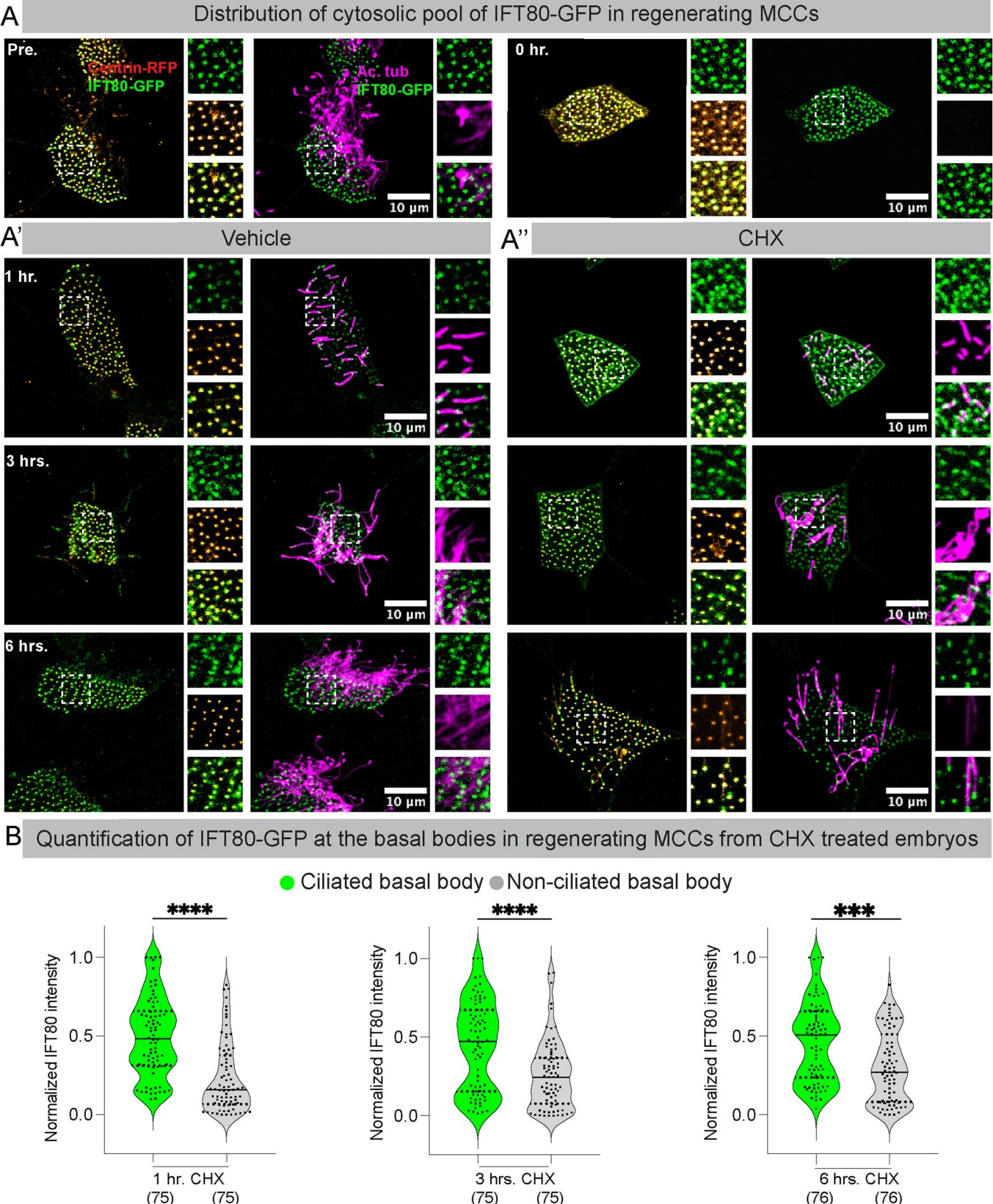

**A** Distribution of cytosolic pool of IFT80-GFP in regenerating MCCs

**A'** Vehicle

**A''** CHX

**B** Quantification of IFT80-GFP at the basal bodies in regenerating MCCs from CHX treated embryos

● Ciliated basal body ● Non-ciliated basal body

◀ **Figure EV4.  Distribution of ciliary precursor pool (IFT80-GFP) in MCCs.**

(A) Embryos injected with IFT80-GFP (green) and Centrin-RFP (orange hot, basal bodies) were deciliated at stage 28 (0 h) and were split into two experiments (DMSO and CHX). (A′, A″) The embryos in both sets regenerated cilia for 6 h. After fixation, the embryos were stained for cilia(magenta). Note that the control MCCs at all time points (1 h, 3 h, and 6 h) have multiple cilia regenerating, and the IFT80-GFP intensity is uniform at every basal body. In contrast, the number of regenerating cilia decreases with time in CHX-treated samples, and the IFT80-GFP is enriched at a few ciliated basal bodies. (B) The intensity of IFT80-GFP associated with ciliated (green) vs. non-ciliated(gray) basal bodies in the same MCC (in CHX-treated samples) at different time points during cilia regeneration. A total of 8–10 basal bodies per MCC (4–5 ciliated and 4–5 non-ciliated) and 5 MCCs were chosen, and the mean gray value was estimated and normalized to the maximum and the minimum values in the set. The value in parenthesis indicates the number of basal bodies analyzed (with and without IFT80-GFP) from 9 embryos from three independent trials. Note the significant difference in the IFT80-GFP signal intensity at ciliated vs. non-ciliated basal bodies at all time points. ****$P < 0.0001$, values on the comparison bar denote $p$ value; Kruskal-Wallis test, followed by Dunn's multiple comparison test.

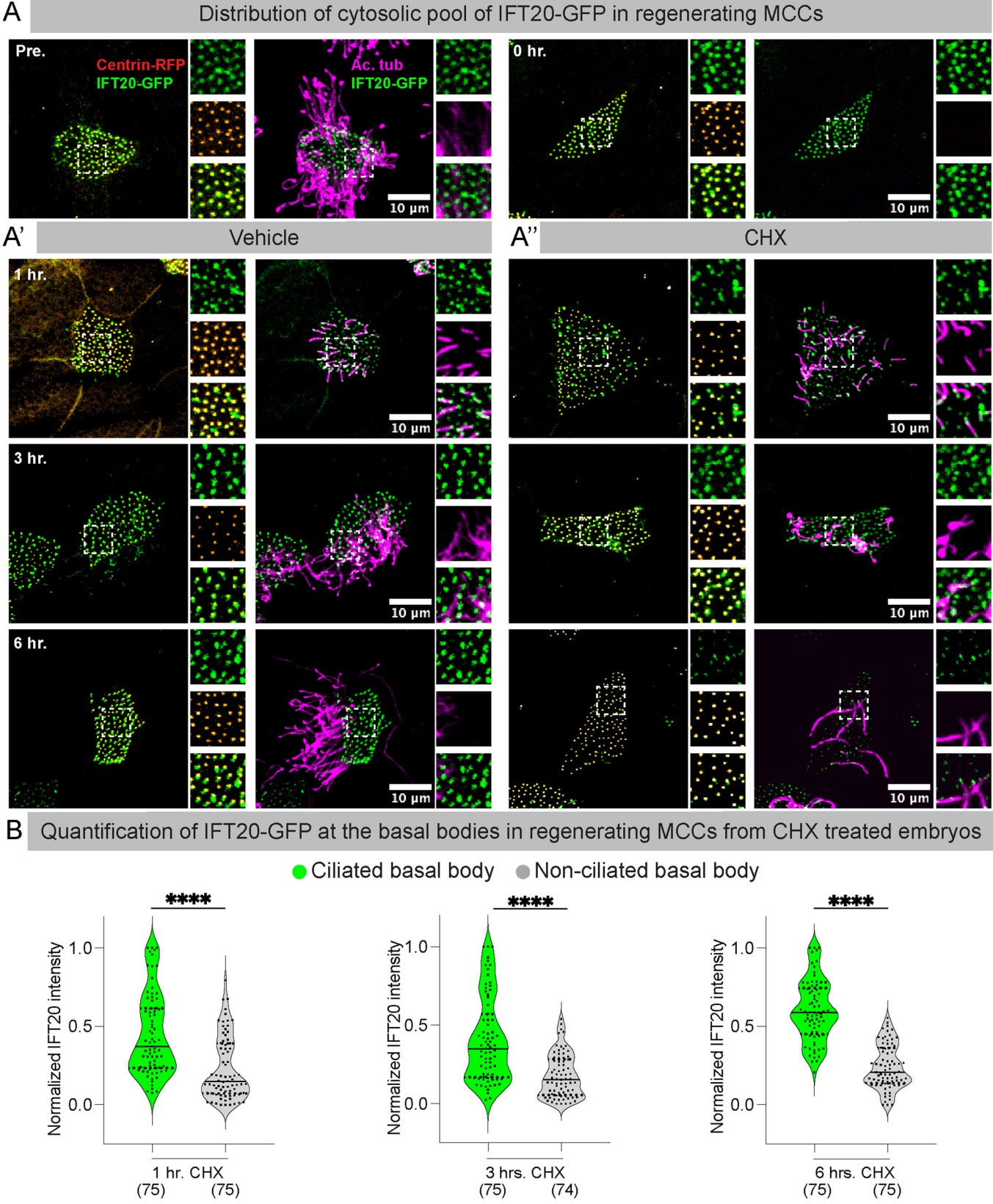

**A** Distribution of cytosolic pool of IFT20-GFP in regenerating MCCs

Pre.
Centrin-RFP
IFT20-GFP
Ac. tub
IFT20-GFP
10 μm

0 hr.
10 μm

**A'** Vehicle

1 hr.
10 μm

3 hr.
10 μm

6 hr.
10 μm

**A''** CHX

10 μm

10 μm

10 μm

**B** Quantification of IFT20-GFP at the basal bodies in regenerating MCCs from CHX treated embryos

● Ciliated basal body    ● Non-ciliated basal body

****
Normalized IFT20 intensity
1 hr. CHX
(75)    (75)

****
Normalized IFT20 intensity
3 hrs. CHX
(75)    (74)

****
Normalized IFT20 intensity
6 hrs. CHX
(75)    (75)

**Figure EV5.   Distribution of ciliary precursor pool (IFT20-GFP) in MCCs.**

(A) Embryos injected with IFT20-GFP (green) and Centrin-RFP (orange hot, basal bodies) were deciliated at stage 28 (0 h) and were split into two experiments (DMSO and CHX). (A′, A″) The embryos in both sets regenerated cilia for 6 h. After fixation, the embryos were stained for cilia(magenta). Note that the control MCCs at all time points (1 h, 3 h, and 6 h) have multiple cilia regenerating, and the IFT20-GFP intensity is uniform at every basal body. In contrast, the number of regenerating cilia decreases with time in CHX-treated samples, and the IFT20-GFP is enriched at a few ciliated basal bodies. (B) The intensity of IFT20-GFP associated with ciliated (green) vs. non-ciliated(gray) basal bodies in the same MCC (in CHX-treated samples) at different time points during cilia regeneration. A total of 8–10 basal bodies per MCC (4–5 ciliated and 4–5 non-ciliated) and 5 MCCs were chosen, and the mean gray value was estimated and normalized to the maximum and the minimum values in the set. The value in parenthesis indicates the number of basal bodies analyzed (with and without IFT20-GFP) from 9 embryos from three independent trials. Note the significant difference in the IFT20-GFP signal intensity at ciliated vs. non-ciliated basal bodies at all time points. ****$P < 0.0001$, Kruskal–Wallis test, followed by Dunn's multiple comparison test.

