## [Peer Review File · EMBO Reports]

Mechanisms of cilia regeneration in *Xenopus* multiciliated epithelium in vivo.

Venkatramanan Rao, Vignesharavind Subramanianbalachandar, Magdalena Magaj, Stefanie Redemann, and Saurabh Kulkarni

Corresponding author(s): Saurabh Kulkarni (sk4xq@virginia.edu)

Review Timeline:

Transfer Date:	12th Aug 24
Editorial Decision:	20th Aug 24
Revision Received:	5th Dec 24
Editorial Decision:	20th Jan 25
Revision Received:	4th Feb 25
Accepted:	18th Feb 25

Editor: Deniz Senyilmaz Tiebe

Transaction Report: This manuscript was transferred to EMBO reports following peer review at Review Commons.

**Review
COMMONS**

Review #1

1. Evidence, reproducibility and clarity:

Evidence, reproducibility and clarity (Required)

In this manuscript entitled "Mechanisms of cilia regeneration in *Xenopus* multiciliated epithelium in vivo", the authors mostly focus on the question, whether TZ (transition zone of cilia) plays an essential role for ciliogenesis during cilia regeneration in multiciliated cells. They used *Xenopus* embryo as a system to examine this question. While cilia regeneration has been actively studied in unicellular green algae, *Chlamydomonas reinhardtii*, the mechanism of cilia regeneration is not known yet. Their approach is to investigate cells after deciliation by calcium shock, based on a TZ protein B9D1, as well as ultrastructure observation using conventional electron microscopy.

The authors observed loss of signal from B9D1 and H-shaped objects, which is typical for TZ, upon deciliation induced by calcium and also during the following re-growth of cilia. Based on these experiments they concluded that TZ formation is not necessary for cilia regeneration in multiciliated cells, differently from *Chlamydomonas*. They further conducted experiments to pursue source of component proteins for re-generation. They compared CHX-treated cells (lacking new protein production) and CHX/MG132 (reduced protein degradation) treated cells to find how the massive amount of protein components upon re-ciliation for multiple cilia will be supplied and regulated. This reviewer found the results of the experiments clearly presented and conducted properly.

The work would have significant impact in the cilia community, if the conclusion is correct. This reviewer, however, has a concern about the authors concluding the presence/absence of TZ, based on only B9D1 and the H-shaped body among nine doublet microtubules. First, is it really established how the structure of *Xenopus* embryo TZ is? While *Chlamydomonas* is well known to have a H-shaped TZ, other species have different form inside the 9+0 doublet, or no feature (Comparison of TZ from various species in Dennis Diener <https://doi.org/10.1016/B978-0-12-822508-0.00007-1>). Fig.2B of this manuscript shows visible densities in the panel "Pre", but it does not look like an H-shape. The tomogram of TZ before deciliation seems clearer (but judging from wavy MTs and membrane in this tomogram, there could be unevenness of embedding and staining), while the tomogram after deciliation is thin and does not cover the entire width. Therefore it is not sure that absence of TZ can be concluded. If the author claims *Xenopus* embryo cilia have a H-shaped TZ, they have to provide multiple micrographs (ideally tomogram or serial section TEM to cover the entire TZ structure) and/or past literature on *Xenopus* embryo TZ. B9D1 is likely a membrane associated protein (according to their deciliation by detergent and mechanical force). This may mean B9D1 is located on or near the membrane, in vicinity to TZ, and thus binds to TZ after the main part of TZ is built. In this case, it is risky to judge presence of TZ based on B9D1. Also in this point, TEM imaging will be helpful to confirm the authors' conclusion.

Their discussion about length/number of cilia and force generated by cilia is interesting, but in the context of this research, this reviewer is skeptical about its value. The calcium induced deciliation is not a physiological phenomena, but an artificial event (please correct if I am wrong). The argument how length and number of cilia are regulated upon deciliation makes sense only in case deciliation happens regularly and the species must optimize themselves to survive.

The argument about possible passway of protein transport to control ciliary number and length (Line408-) seems, although it is an interesting topic in general, not suitable in this manuscript. For this reviewer's view, it is relatively straightforward to interpret the result of cilia number/length under normal growth, without new protein expression (CHX), with protein degradation blocked. Cilia will extend when components are provided. Growth will slow down when it is exhausted. Existing cilia start degrading, when they lack proteins, which are necessary for turn-over. With the current experimental output, there is no point to describe redistribution of proteins.

****Minor points:****

Line65: do they mean "selected few basal bodies"?

Line73: extracellular flow is not limited to developmental system.

Line124: alpha-tubulin signal and SEM image

Line139: Could you define explicitly the two hypotheses?

Line164: 10,31-33 are not suitable citation for the location of calcium induced deciliation in Chlamydomonas. cite Sanders and Salisbury JCB 108, 1751

Line181: Later -> latter

Line195: by mechanical shearing, B9D1 remained with cilia. They concluded that TZ stays with the axoneme by deciliation. How can they exclude the possibility that mechanical separation works differently from calcium shock?

Line214: 1.33uM -> 1.33um

2. Significance:

Significance (Required)

The work would have significant impact in the cilia community, if the conclusion is correct. Their discussion about length/number of cilia and force generated by cilia is interesting, but in the context of this research, this reviewer is skeptical about its value.

3. How much time do you estimate the authors will need to complete the suggested revisions:

Estimated time to Complete Revisions (Required)

(Decision Recommendation)

Between 3 and 6 months

4. Review Commons values the work of reviewers and encourages them to get credit for their work. Select 'Yes' below to register your reviewing activity at Web of Science Reviewer

Recognition Service (formerly Publons); note that the content of your review will not be visible on Web of Science.

Yes

Review #2

1. Evidence, reproducibility and clarity:

Evidence, reproducibility and clarity (Required)

****Summary****

This manuscript investigates how cilia regenerate in multi-ciliated cells. The authors have exploited an original multi-ciliated cell system derived from the *Xenopus* embryonic cap and use chemical and mechanical deciliation to understand the different steps of cilia regeneration. In this model, they show that cilia are excised just above the BB and below the ciliary transition zone. Their results indicate that during ciliary regeneration, axoneme reassembly precedes TZ formation and that ciliary reassembly relies on de novo protein synthesis. In the context of limited protein synthesis, cells regenerate fewer cilia, but of almost the same size as control cells, suggesting the existence of a cell control system to maximise force generation. Mathematical modelling of the forces exerted by defined numbers of cilia of different lengths supports this hypothesis.

****Major comments****

Overall, the results are well presented and allow strong conclusions to be drawn. The results are based on both immunofluorescence studies and EM analysis. To support their observations that cilia length is favored over cilia number under conditions of limiting ciliary precursor availability, the authors use a mathematical model that leads to the conclusion that force generation is optimized by increasing cilia length. This is a convincing conclusion, and in agreement with other comparable modeling studies performed in the field. It would be fascinating to be able to measure the flow parameters at the cell surface during cilia regeneration to see whether this regeneration actually leads to an increase in the overall flow or force generated by the cilia. But as the authors explain, this is probably a difficult experiment to carry out and appears to be optional in the context of this study.

The authors are apparently only able to detect a single TZ protein, B9D1, to follow the fate of the TZ during the deciliation and reciliation process. In some ways, this provides an incomplete demonstration that all the TZ is indeed removed during deciliation, although this is supported by EM observations. It also provides a limited understanding of the time course of TZ re-formation during reciliation. Given the limitations of antibody availability, could it be possible to express tagged

proteins in the animal cap system to track more TZ proteins? In particular, would it be possible to track for example Cby and NPHP proteins. What is the behavior of Cep290? This would greatly reinforce the conclusions on the molecular reorganisation of the TZ after deciliation and during cilia regeneration.

****Minor comments****

Figure 4: The images are poorly defined and it is difficult to distinguish individual basal bodies and cilia. It is therefore not clear how the authors can confidently quantify the number of basal bodies in each condition to construct the graph at the bottom of the figure. In addition, it would be interesting to label the basal body with a centriolar marker to better define the basal body.

Figure 5: not clear why the graph on the lower right does not include the control at 3 and 6 hrs? Is it because the number is too high and difficult to quantify?

References: I would like to draw the authors' attention to studies of deciliation in *Paramecia* that could be cited in the introduction or discussion of the conservation of this pathway through evolution.

2. Significance:

Significance (Required)

The mechanisms of deciliation and re-ciliation have mostly been studied in protozoa (*Chlamydomonas*, *Paramecia*) or in primary ciliated cell cultures. Only a few studies have described deciliation in multiciliated cells, such as sea urchins, or physiological deciliation in the oviduct. The *Xenopus* deciliation system described here has already been used to determine the dynamics of IFT proteins during ciliogenesis or to define the ciliary proteome. In this study, the authors go one step further by describing more precisely which part of the cilium is shed upon induction of deciliation and the dynamics of the recruitment of the Tip and of the TZ proteins.

This study provides a completely new perspective on the deciliation process:

1. the authors show that, contrary to what is generally accepted from protozoan studies, the deciliation process, in *Xenopus* multiciliated cells, expels the TZ, leaving only the basal body in the cell;
2. While ciliogenesis is described in various models to begin with the formation of the TZ, in this *Xenopus* system the TZ matures after the onset of axonemal elongation, calling into question the precise function of the TZ in axonemal elongation. The observations could be further strengthened by analyzing more TZ proteins to better understand the time course of events involved in the deciliation-re-ciliation program.

The protocol used to deciliate *Xenopus* multiciliated cells has been described in previous manuscripts. Its use here reveals striking differences in the deciliation-reconciliation pathways from what is known in the field. It provides new conceptual perspectives for researchers working on the basic mechanisms of ciliogenesis. Note that, as a geneticist and specialist in ciliogenesis using

various model organisms, I am not fully competent to critically evaluate the mathematical models developed in this study.

3. How much time do you estimate the authors will need to complete the suggested revisions:

Estimated time to Complete Revisions (Required)

(Decision Recommendation)

Between 3 and 6 months

4. Review Commons values the work of reviewers and encourages them to get credit for their work. Select 'Yes' below to register your reviewing activity at Web of Science Reviewer Recognition Service (formerly Publons); note that the content of your review will not be visible on Web of Science.

Yes

Review #3

1. Evidence, reproducibility and clarity:

Evidence, reproducibility and clarity (Required)

The manuscript by Rao et al. focuses on determining the mechanism of cilia regeneration using *Xenopus* mucociliary epithelium. The authors employ a simple yet powerful approach to trigger deciliation of multiciliated cells, enabling them to study the mechanism of cilia regeneration. This research has a significant impact on the field of cilia biology and enhances our understanding of ciliopathies. Through detailed cell biological methodologies, the authors obtained intriguing results, including the finding that deciliation removes the transition zone and that cilia repair precedes the transition zone assembly. Additionally, the authors demonstrate that IFT proteins involved in cilia construction concentrate at selected basal bodies. Although there are open questions that the authors also highlight, this manuscript provides solid, pioneering insights into the process of cilia regeneration in vivo.

2. Significance:

Significance (Required)

The manuscript characterizes the mechanism of cilia regeneration, providing new insights into processes that could be harnessed to restore ciliary function in patients suffering from chronic respiratory diseases.

3. How much time do you estimate the authors will need to complete the suggested revisions:

Estimated time to Complete Revisions (Required)

(Decision Recommendation)

Cannot tell / Not applicable

No

Revision Plan

Manuscript number: RC-2024-02512

Corresponding author(s): Saurabh S. Kulkarni

[The “revision plan” should delineate the revisions that authors intend to carry out in response to the points raised by the referees. It also provides the authors with the opportunity to explain their view of the paper and of the referee reports.]

The document is important for the editors of affiliate journals when they make a first decision on the transferred manuscript. It will also be useful to readers of the reprint and help them to obtain a balanced view of the paper.

*If you wish to submit a full revision, please use our "Full Revision" template. **It is important to use the appropriate template to clearly inform the editors of your intentions.**]*

1. General Statements [optional]

First, we would like to thank the reviewers for reviewing and commenting on our manuscript. All three reviewers have lauded our work as highly significant and novel. Specifically, Reviewer 1 stated that “The work would have significant impact in the cilia community”. Reviewer 2 states that “This study provides a completely new perspective on the deciliation process”, and “reveals striking differences in the deciliation-reconciliation pathways from what is known in the field. It provides new conceptual perspectives for researchers working on the basic mechanisms of ciliogenesis”. Reviewer 3 states that “This research has a significant impact on the field of cilia biology and enhances our understanding of ciliopathies”, and “Although there are open questions that the authors also highlight, this manuscript provides solid, pioneering insights into the process of cilia regeneration in vivo.”

2. Description of the planned revisions

Insert here a point-by-point reply that explains what revisions, additional experimentations and analyses are planned to address the points raised by the referees.

Reviewer 1:

1. The work would have significant impact in the cilia community, if the conclusion is correct. This reviewer, however, has a concern about the authors concluding the presence/absence of TZ, based on only B9D1 and the H-shaped body among nine doublet microtubules. First, is it really established how the structure of *Xenopus* embryo TZ is? While *Chlamydomonas* is well known to have a H-shaped TZ, other species have different form inside the 9+0 doublet, or no feature (Comparison of TZ from various species in Dennis Diener <https://doi.org/10.1016/B978-0-12-822508-0.00007-1>). Fig.2B of this manuscript shows visible densities in the panel "Pre", but it does not look like an H-shape. The tomogram of TZ before deciliation seems clearer (but judging from wavy MTs and membrane in this tomogram, there could be unevenness of embedding and staining), while the tomogram after deciliation is thin and does not cover the entire width. Therefore it is not sure that absence of TZ can be

Revision Plan

concluded. If the author claims *Xenopus* embryo cilia have a H-shaped TZ, they have to provide multiple micrographs (ideally tomogram or serial section TEM to cover the entire TZ structure) and/or past literature on *Xenopus* embryo TZ. B9D1 is likely a membrane associated protein (according to their deciliation by detergent and mechanical force). This may mean B9D1 is located on or near the membrane, in vicinity to TZ, and thus binds to TZ after the main part of TZ is built. In this case, it is risky to judge presence of TZ based on B9D1. Also in this point, TEM imaging will be helpful to confirm the authors' conclusion.

RESPONSE: We appreciate the reviewer's thoughtful comments on the loss of TZ upon deciliation and its absence during the initial regeneration period. The reviewer is right in their assessment that the TZ of *Xenopus* cilia has not been well defined before in any manuscript. We want the reviewer to consider that our goal was not to define the TZ in *Xenopus* but to study deciliation and how cilia regenerate in a vertebrate model system for the first time. We unexpectedly discovered that cilia are deciliated distal to the basal body at the plasma membrane, and the "H-shaped structure," similar to TZ, was also removed and did not come back for first hour during regeneration. Given this surprising observation, we felt obliged to study and explain our results. To that end, we explored different resources (antibodies and markers of TZ) and different methods over 6 years trying to define TZ in *Xenopus*.

Our conclusion about the TZ structure came from multiple lines of evidence from our experiments and published literature, including the similarity in structure compared to other organisms and its physical location in the cilium. Specifically, 1) In a review of the basal bodies, Mitchell indirectly suggested that the electron-dense "H-shaped" structure could be a TZ in *Xenopus*. 2) The electron-dense "H" shaped structure in *Chlamydomonas* is similar, if not identical, to that shown in *Xenopus* cilia. 3) The physical location of TZ is always shown to be distal to the basal body and transition fibers (except in clubmoss *Phylloglossum*) while proximal to the central pair. The electron-dense "H-shaped" structure in *Xenopus* fulfills these criteria, suggesting that this structure is the TZ in *Xenopus*. 4) The TZ bonafide protein B9D1 is localized distal to Chibby, which labels the distal end of the basal body, suggesting that the TZ is localized distal to the basal body. Moreover, the loss of an "H-shaped" structure determined using TEM and tomograms corresponds to the loss of the B9D1 signal, further strengthening the conclusion that the H-shaped structure is the TZ.

We will include serial sectioning and imaging of multiple *Xenopus* cilia in control and 0hr (after deciliation) to address this reviewer's concerns further. Our preliminary data has suggested that the ciliary membrane is tightened around this electron-dense structure, similar to what has been shown before for other organisms like *Chlamydomonas*. and thus boosts our confidence that this structure likely corresponds to the TZ in *Xenopus*.

The reviewer has raised a concern that "the tomogram after deciliation is thin and does not cover the entire width. Therefore, it is not sure that absence of TZ can be concluded". We note that even if the tomograms do not go through the entire cilium (supplementary videos 2 and 3), it does go through more than the center of cilium as seen by the presence of central pair microtubules and we can observe that the electron-dense "H-shaped" structure is not present in these cilia. Further,

Revision Plan

in the supplementary videos 5 and 6, even if the tomogram again only covers half of the cilia, we can see the presence of the structure, confirming that our tomograms can demonstrate the presence or absence of the H-shaped structure confidently. We have also provided TEM sections in addition to the tomograms to show the same result.

The Reviewer has commented that “B9D1 is located on or near the membrane, in vicinity to TZ, and thus binds to TZ after the main part of TZ is built”. This reviewer is correct in their assessment. This is why we argue that the presence or absence of B9D1 may be a good marker for understanding the presence or absence of TZ assembly.

TIMELINE: We are performing additional serial TEM in the control and deciliated (0hr.) embryos to address the reviewer’s concern. We will need 1 month to finish these experiments.

2. Their discussion about length/number of cilia and force generated by cilia is interesting, but in the context of this research, this reviewer is skeptical about its value. The calcium induced deciliation is not a physiological phenomenon, but an artificial event (please correct if I am wrong). The argument how length and number of cilia are regulated upon deciliation makes sense only in case deciliation happens regularly and the species must optimize themselves to survive. The argument about possible passway of protein transport to control ciliary number and length (Line408-) seems, although it is an interesting topic in general, not suitable in this manuscript. For this reviewer's view, it is relatively straightforward to interpret the result of cilia number/length under normal growth, without new protein expression (CHX), with protein degradation blocked. Cilia will extend when components are provided. Growth will slow down when it is exhausted. Existing cilia start degrading, when they lack proteins, which are necessary for turn-over. With the current experimental output, there is no point to describe redistribution of proteins.

RESPONSE: We appreciate the reviewer’s comment; however, we would like to argue that different methods of deciliation have been used in different model systems, such as Chlamydomonas, to study cilia regeneration. Although this reviewer may not find some of the experiments and conclusions appropriate for this manuscript, other research groups have found these results interesting. For example, reviewer 2 states, “To support their observations that cilia length is favored over cilia number under conditions of limiting ciliary precursor availability, the authors use a mathematical model that leads to the conclusion that force generation is optimized by increasing cilia length. This is a convincing conclusion and in agreement with other comparable modeling studies performed in the field.” We have already had great discussions about these results with many cilia researchers at multiple conferences. Therefore, we prefer to keep these experiments and results in the manuscript and let readers come to their own conclusions about their importance.

Minor points:

Line65: do they mean "selected few basal bodies"? – we have removed the word “select”

Line73: extracellular flow is not limited to developmental system. – we have altered the statement to add “growth, development and homeostasis”

Revision Plan

Line124: alpha-tubulin signal and SEM image – we have added “and scanning electron microscopy (SEM)”

Line139: Could you define explicitly the two hypotheses? – Now, we have reworded the sentences to clarify the two hypotheses. “Therefore, we considered two hypotheses: First, *Xenopus* MCCs regenerate cilia or second, *Xenopus* depend on stem cell-based replacement of damaged MCCs.”

Line164: 10,31-33 are not suitable citation for the location of calcium induced deciliation in *Chlamydomonas*. cite Sanders and Salisbury JCB 108, 1751 – We have changed the citation.

Line181: Later -> latter – We have changed the text.

Line195: by mechanical shearing, B9D1 remained with cilia. They concluded that TZ stays with the axoneme by deciliation. How can they exclude the possibility that mechanical separation works differently from calcium shock? – We do not intend to claim that both calcium-based and mechanical ripping of cilia from cells adopt the same deciliation mechanism, and we have mentioned in line 193 that ‘we adopted an alternative approach of mechanical deciliation’. Using these two methods as complimentary to each other, our aim was to show that TZ is lost by both ciliation methods. For the calcium method, because the membrane is ripped with detergent, we show the loss of TZ by examining the MCCs devoid of cilia. In the mechanical deciliation protocol, since no detergent is involved, we can examine cilia that are likely to have intact membranes and thus maintain a B9D1 signal.

Line214: 1.33uM -> 1.33um - We have made these changes to the text.

RESPONSE: All the minor points in the manuscript are addressed.

Reviewer 2

1. Overall, the results are well presented and allow strong conclusions to be drawn. The results are based on both immunofluorescence studies and EM analysis. To support their observations that cilia length is favored over cilia number under conditions of limiting ciliary precursor availability, the authors use a mathematical model that leads to the conclusion that force generation is optimized by increasing cilia length. This is a convincing conclusion, and in agreement with other comparable modeling studies performed in the field. It would be fascinating to be able to measure the flow parameters at the cell surface during cilia regeneration to see whether this regeneration actually leads to an increase in the overall flow or force generated by the cilia. ***But as the authors explain, this is probably a difficult experiment to carry out and appears to be optional in the context of this study.***

RESPONSE: We thank the reviewer for recognizing and stating that “the results are well presented and allow strong conclusions to be drawn”. We also want to sincerely thank the reviewer for understanding the technical difficulties in performing these experiments.

2. The authors are apparently only able to detect a single TZ protein, B9D1, to follow the fate of the TZ during the deciliation and reciliation process. In some ways, this provides an incomplete demonstration that all the TZ is indeed removed during deciliation, although this is supported by EM observations. It also provides a limited understanding of the time course of TZ re-formation during reciliation. Given the limitations of antibody availability, could it be possible to express tagged proteins in the animal cap system to track more TZ proteins? In particular, would it be possible to track for example Cby and NPHP proteins. What is the

Revision Plan

behavior of Cep290? This would greatly reinforce the conclusions on the molecular reorganisation of the TZ after deciliation and during cilia regeneration.

RESPONSE: We appreciate this reviewer's brilliant questions on understanding the time course of TZ re-formation during reciliation. When we started this project and observed that TZ was lost upon deciliation in our preliminary TEM experiment, our first goal was to confirm this outstanding result. Thus, we did more TEMs and EM tomography, used bonafide TZ protein B9D1 to label the structure, and observed its loss upon deciliation. Taken together, we feel highly confident that TZ is lost upon deciliation. To address this reviewer's concerns, we will performing additional serial TEMs to confirm the loss of TZ after deciliation.

Our next goal was to understand what the reviewer has mentioned, the TZ assembly time course. We started with TEMs at different time points and again saw a surprising result: TZ assembly was delayed compared to cilia axoneme. We were driven by this question of understanding how cilia "put together" the complex structure of TZ structurally and molecularly using EM and fluorescence data. We first attempted a few antibodies, including B9D1, CEP290, MKS5, and NPHP4, to localize to the TZ in the *Xenopus* cilia. Despite our efforts with different fixation strategies, only B9D1 appeared to localize to the TZ, whereas others did not give any signal or localized at the basal body. Next, we tried localizing TMEM216, TMEM67, and NPHP4 using fluorescent tags, but we again found the same result: they localized to the basal body but not at the TZ. We are perplexed by this result and are pursuing the reasons behind them. However, these experiments are out of the scope of this paper. We want to note that we have used Chibby in our experiments and that it is not lost upon deciliation (Fig S1). This is because Chibby is a distal transition fiber protein (distal end of basal body) and does not extend up to the transition zone.

TIMELINE: To address the reviewer's concern, we are performing additional serial TEM in the control and deciliated (0hr.) embryos. We will attempt to localize CEP290-GFP, requiring approximately 1 month to finish the experiment. However, we would like to note that we cannot guarantee that this experiment will work, as similar experiments with other TZ markers have failed before.

Minor comments

3. Figure 4: The images are poorly defined, and it is difficult to distinguish individual basal bodies and cilia. Therefore, it is not clear how the authors can confidently quantify the number of basal bodies in each condition to construct the graph at the bottom of the figure. In addition, it would be interesting to label the basal body with a centriolar marker to better define it. - Figure 4 labels the Transition Zone protein B9D1 and cilia marker acetylated tubulin and not basal bodies. The graph represents the number of cells with the presence or absence of elongated B9d1 signal.

Revision Plan

4. Figure 5: not clear why the graph on the lower right does not include the control at 3 and 6 hrs? Is it because the number is too high and difficult to quantify? – Yes, the reviewer is right. Cilia become too long and too many to quantify their number reliably.
5. References: I would like to draw the authors' attention to studies of deciliation in Paramecia that could be cited in the introduction or discussion of the conservation of this pathway through evolution. – We have added multiple references to paramecia throughout the manuscript. Specifically, we mention that deciliation and regeneration in unicellular models like paramecia have added to our understanding of ciliogenesis. Line 102 “While it is important to remember that regeneration of cilia may not be identical to *de novo* assembly, cilia regeneration studies in *Chlamydomonas reinhardtii*, *Paramecium* and *Tetrahymena* etc., have provided significant insights into ciliogenesis, e.g., cargo transport, the presence of precursor pool, regulation of ciliary gene expression.^{18,23–26}”. Further, we also added the reference to paramecia in results, line 164 “Next, we determined the location where the deciliation treatment severed cilia. Unicellular models such as *Chlamydomonas*, *Paramecium* and *Tetrahymena* lose cilia distal to the TZ and below the central pair (CP) microtubules³³.”. We also add discussion on the importance of TZ in paramecia, line 203 “Interestingly in *Paramecium* also a unicellular multiciliated cell, displays constant shedding of cilia when TZ proteins are depleted.²⁵”. These statements have been supported by the following studies that are now cited in the manuscript: Machemer and Ogura 1979 Journal of Cell Physiology (10.1113/jphysiol.1979.sp012990) and Gogenddeau et al., Plos Biology (10.1371/journal.pbio.3000640).

RESPONSE: All the minor points in the manuscript are addressed.

Reviewer 3

Reviewer 3 did not have any major or minor comments.

3. Description of the revisions that have already been incorporated in the transferred manuscript

Please insert a point-by-point reply describing the revisions that were already carried out and included in the transferred manuscript. If no revisions have been carried out yet, please leave this section empty.

4. Description of analyses that authors prefer not to carry out

Please include a point-by-point response explaining why some of the requested data or additional analyses might not be necessary or cannot be provided within the scope of a revision. This can be due to time or resource limitations or in case of disagreement about the necessity of such additional data given the scope of the study. Please leave empty if not applicable.

RESPONSE: The response above explains any experiment we cannot perform. Further, we disagree with Reviewer 1's second comment to remove some of the results from the manuscript. We have added the reasoning for our disagreement above in the response.

Dear Dr. Kulkarni,

Thank you for submitting your manuscript to EMBO Reports, which was previously reviewed at Review Commons.

Referees express interest in the study investigating the mechanisms of re-ciliation in *Xenopus* multiciliated epithelium. However, they also raise concerns that need to be addressed to consider publication in EMBO Reports. In particular, referees #2 and #3 requests additional support to the conclusion that formation of axoneme precedes TZ formation, which needs to be addressed satisfactorily for publication here.

Having looked at all documents, we would like to invite you to submit a revised manuscript as in your revision plan. Please revise your manuscript with the understanding that the referee concerns (as in their reports) must be fully addressed and their suggestions taken on board. Please address all referee concerns in a complete point-by-point response. Acceptance of the manuscript will depend on a positive outcome of a second round of review. It is EMBO reports policy to allow a single round of major experimental revision only and acceptance or rejection of the manuscript will therefore depend on the completeness of your responses included in the next, final version of the manuscript.

We realize that it is difficult to revise to a specific deadline. In the interest of protecting the conceptual advance provided by the work, we recommend a revision within 3 months. Please discuss the revision progress ahead of this time with me if you require more time to complete the revisions, or if you have questions or comments regarding the revision (also by video chat).

1. A data availability section providing access to data deposited in public databases is missing (where applicable).
2. Your manuscript contains statistics and error bars based on $n=2$. Please use scatter plots in these cases.

You can submit the revision either as a Scientific Report or as a Research Article. For Scientific Reports, the revised manuscript can contain up to 5 main figures and 5 Expanded View figures, and it should not exceed 27000 characters. If the revision leads to a manuscript with more than 5 main figures it will be published as a Research Article. In this case the Results and Discussion section should be separate. If a Scientific Report is submitted, these sections have to be combined. This will help to shorten the manuscript text by eliminating some redundancy that is inevitable when discussing the same experiments twice. In either case, all materials and methods should be included in the main manuscript file.

4) a .docx formatted letter INCLUDING the reviewers' reports and your detailed point-by-point responses to their comments. As part of the EMBO publication's Transparent Editorial Process, EMBO reports publishes online a Review Process File (RPF) to accompany accepted manuscripts. This File will be published in conjunction with your paper and will include the referee reports, your point-by-point response and all pertinent correspondence relating to the manuscript.

<https://www.embopress.org/page/journal/14693178/authorguide#transparentprocess>

You are able to opt out of this by letting the editorial office know (emboreports@embo.org). If you do opt out, the Review Process File link will point to the following statement: "No Review Process File is available with this article, as the authors have

chosen not to make the review process public in this case."

5) a complete author checklist, which you can download from our author guidelines

<https://www.embopress.org/page/journal/14693178/authorguide>. Please insert information in the checklist that is also reflected in the manuscript. The completed author checklist will also be part of the RPF.

6) Please note that all corresponding authors are required to supply an ORCID ID for their name upon submission of a revised manuscript (). Please find instructions on how to link your ORCID ID to your account in our manuscript tracking system in our Author guidelines

Additional information on source data and instruction on how to label the files are available:

<https://www.embopress.org/page/journal/14693178/authorguide#sourcedata>

9) Our journal encourages inclusion of *data citations in the reference list* to directly cite datasets that were re-used and obtained from public databases. Data citations in the article text are distinct from normal bibliographical citations and should directly link to the database records from which the data can be accessed. In the main text, data citations are formatted as follows: "Data ref: Smith et al, 2001" or "Data ref: NCBI Sequence Read Archive PRJNA342805, 2017". In the Reference list, data citations must be labeled with "[DATASET]". A data reference must provide the database name, accession number/identifiers and a resolvable link to the landing page from which the data can be accessed at the end of the reference. Further instructions are available at <http://www.embopress.org/page/journal/14693178/authorguide#referencesformat>

10) Regarding data quantification (see Figure Legends:

<https://www.embopress.org/page/journal/14693178/authorguide#figureformat>)

12) Please also note our reference format:

13) All Materials and Methods need to be described in the main text using our 'Structured Methods' format, which is required for all research articles. According to this format, the Methods section includes a Reagents and Tools Table (listing key reagents, experimental models, software and relevant equipment and including their sources and relevant identifiers) followed by a Methods and Protocols section describing the methods using a step-by-step protocol format. The aim is to facilitate adoption of the methodologies across labs. More information on how to adhere to this format as well as a downloadable template (.docx) for the Reagents and Tools Table can be found in our author guidelines:
<https://www.embopress.org/page/journal/14693178/authorguide#structuredmethods>.

An example of a Method paper with Structured Methods can be found here:
<https://www.embopress.org/doi/10.15252/msb.20178071>.

I look forward to seeing a revised version of your manuscript when it is ready. Please let me know if you have questions or comments regarding the revision.

Kind regards,

Deniz Senyilmaz Tiebe

Deniz Senyilmaz Tiebe, PhD
Senior Scientific Editor
EMBO Reports

We want to thank the reviewers for their service. The reviewers have identified numerous strengths in the submitted manuscript, including but not limited to “The work would have a significant impact in the cilia community”, “results of the experiments clearly presented and conducted properly”, and the study provides new perspectives on deciliation and regeneration in vertebrates, the importance of TZ in cilia regeneration, and the basic mechanisms involved in ciliogenesis. Despite these merits, the reviewers expressed few concerns. We have addressed all concerns in response to their constructive feedback, enhancing the manuscript's impact.

REVISIONS

Reviewer 1:

1. The work would have significant impact in the cilia community, if the conclusion is correct. This reviewer, however, has a concern about the authors concluding the presence/absence of TZ, based on only B9D1 and the H-shaped body among nine doublet microtubules. First, is it really established how the structure of *Xenopus* embryo TZ is? While *Chlamydomonas* is well known to have a H-shaped TZ, other species have different form inside the 9+0 doublet, or no feature (Comparison of TZ from various species in Dennis Diener <https://doi.org/10.1016/B978-0-12-822508-0.00007-1>). Fig.2B of this manuscript shows visible densities in the panel "Pre", but it does not look like an H-shape. The tomogram of TZ before deciliation seems clearer (but judging from wavy MTs and membrane in this tomogram, there could be unevenness of embedding and staining), while the tomogram after deciliation is thin and does not cover the entire width. Therefore it is not sure that absence of TZ can be concluded. If the author claims *Xenopus* embryo cilia have a H-shaped TZ, they have to provide multiple micrographs (ideally tomogram or serial section TEM to cover the entire TZ structure) and/or past literature on *Xenopus* embryo TZ. B9D1 is likely a membrane associated protein (according to their deciliation by detergent and mechanical force). This may mean B9D1 is located on or near the membrane, in vicinity to TZ, and thus binds to TZ after the main part of TZ is built. In this case, it is risky to judge presence of TZ based on B9D1. Also in this point, TEM imaging will be helpful to confirm the authors' conclusion.

RESPONSE: We appreciate the reviewer's thoughtful comments on the loss of TZ upon deciliation and its absence during the initial regeneration period. The reviewer is right in their assessment that the TZ of *Xenopus* cilia has not been well defined before in any manuscript. We want the reviewer to consider that our goal was not to define the TZ in *Xenopus* but to study deciliation and how cilia regenerate in a vertebrate model system for the first time. We unexpectedly discovered that cilia are deciliated distal to the basal body at the plasma membrane, and the “H-shaped structure,” similar to TZ, was also removed and did not come back for first hour during regeneration. Given this surprising observation, we felt obliged to study and explain our results. To that end, we explored different resources (antibodies and markers of TZ) and different methods over 6 years trying to define TZ in *Xenopus*.

Our conclusion about the TZ structure came from multiple lines of evidence from our experiments and published literature, including the similarity in structure compared to other organisms and its physical location in the cilium. Specifically, 1) In a review of the basal bodies, Brian Mitchell indirectly suggested that the electron-dense “H-shaped” structure could be a TZ in *Xenopus*. 2) The electron-dense “H” shaped structure in *Chlamydomonas* is similar, if not identical, to that shown in *Xenopus* cilia. 3) The physical location of TZ is always shown to be distal to the basal body and transition fibers (except in clubmoss *Phylloglossum*) while proximal to the central pair. The electron-dense “H-shaped” structure in *Xenopus* fulfills these criteria, suggesting that this structure is the TZ in *Xenopus*. 4) The TZ bonafide protein B9D1 is localized distal to Chibby, which labels the distal end of the basal body, suggesting that the TZ is localized distal to the basal body. Moreover, the loss of an “H-shaped” structure determined using TEM and tomograms corresponds to the loss of the B9D1 signal, further strengthening the conclusion that the H-shaped structure is the TZ.

The reviewer has raised a concern that “the tomogram after deciliation is thin and does not cover the entire width. Therefore, it is not sure that the absence of TZ can be concluded. We would like to note that even if the tomograms do not go through the entire cilium, the sections are ~200nm thick, compared to cilia diameter (~250-300nm), and thus, in most cases, include >50% of ciliary axoneme width as seen by the presence of central pair microtubules. To further address the concern, **we have added 8 new tomograms as extended videos to support our data (Movies EV2-14)**. Some of these tomograms clearly show the complete absence of ciliary axoneme, including TZ at some basal bodies at 20min and 1 hr. Other tomograms demonstrate the

TZ assembly at various stages during regeneration. We have also provided 2 TEM sections at pre-deciliation and post-deciliation (0 min. timepoint) in addition to the tomograms to show the same result (Fig. EV2E-2F).

The Reviewer has commented that “B9D1 is located on or near the membrane, in vicinity to TZ, and thus binds to TZ after the main part of TZ is built”. This reviewer is correct in their assessment. This is why we argue that the presence or absence of B9D1 may be a good marker for understanding the presence or absence of TZ assembly.

2. Their discussion about length/number of cilia and force generated by cilia is interesting, but in the context of this research, this reviewer is skeptical about its value. The calcium induced deciliation is not a physiological phenomenon, but an artificial event (please correct if I am wrong). The argument how length and number of cilia are regulated upon deciliation makes sense only in case deciliation happens regularly and the species must optimize themselves to survive. The argument about possible passway of protein transport to control ciliary number and length (Line408-) seems, although it is an interesting topic in general, not suitable in this manuscript. For this reviewer's view, it is relatively straightforward to interpret the result of cilia number/length under normal growth, without new protein expression (CHX), with protein degradation blocked. Cilia will extend when components are provided. Growth will slow down when it is exhausted. Existing cilia start degrading, when they lack proteins, which are necessary for turn-over. With the current experimental output, there is no point to describe redistribution of proteins.

RESPONSE: We appreciate the reviewer's comment; however, we would like to argue that different methods of deciliation have been used in different model systems, such as *Chlamydomonas*, to study cilia regeneration. Although this reviewer may not find some of the experiments and conclusions appropriate for this manuscript, other research groups have found these results interesting. For example, reviewer 2 states, “To support their observations that cilia length is favored over cilia number under conditions of limiting ciliary precursor availability, the authors use a mathematical model that leads to the conclusion that force generation is optimized by increasing cilia length. This is a convincing conclusion and in agreement with other comparable modeling studies performed in the field.” We have already had great discussions about these results with many cilia researchers at multiple conferences. Therefore, we prefer to keep these experiments and results in the manuscript and let readers come to their own conclusions about their importance.

Minor points:

Line 65: do they mean "selected few basal bodies"? – we have removed the word “select” (now line 64)

Line 73: extracellular flow is not limited to developmental system. – we have altered the statement to add “growth, development and homeostasis” (now line 72).

Line 124: alpha-tubulin signal and SEM image – we have added “and scanning electron microscopy (SEM)”, now line 132.

Line 139: Could you define explicitly the two hypotheses? – Now, we have reworded the sentences to clarify the two hypotheses. “Therefore, we considered two hypotheses: First, *Xenopus* MCCs regenerate cilia or second, *Xenopus* depend on stem cell-based replacement of damaged MCCs.”, now line 145-147.

Line 164: 10,31-33 are not suitable citation for the location of calcium induced deciliation in *Chlamydomonas*. cite Sanders and Salisbury JCB 108, 1751 – We have changed the citation, now line 174.

Line 181: Later -> latter – We have changed the text. Now in line 198.

Line 195: by mechanical shearing, B9D1 remained with cilia. They concluded that TZ stays with the axoneme by deciliation. How can they exclude the possibility that mechanical separation works differently from calcium shock? – We do not intend to claim that both calcium-based and mechanical ripping of cilia from cells adopt the same deciliation mechanism, and we have mentioned in line 208 that ‘we adopted an alternative approach of mechanical deciliation’. Using these two methods as complimentary to each other, our aim was to show that TZ is lost by both deciliation methods. For the calcium method, because the membrane is ripped with detergent, we show the loss of TZ by examining the MCCs devoid of cilia. In the mechanical deciliation protocol, since no detergent is involved, we can examine cilia that are likely to have intact membranes and thus maintain a B9D1 signal.

Line 214: 1.33uM -> 1.33um - We have made these changes to the text, now line 235

RESPONSE: All the minor points in the manuscript are addressed.

Reviewer 2

1. Overall, the results are well presented and allow strong conclusions to be drawn. The results are based on both immunofluorescence studies and EM analysis. To support their observations that cilia length is favored over cilia number under conditions of limiting ciliary precursor availability, the authors use a mathematical model that leads to the conclusion that force generation is optimized by increasing cilia length. This is a convincing conclusion, and in agreement with other comparable modeling studies performed in the field. It would be fascinating to be able to measure the flow parameters at the cell surface during cilia regeneration to see whether this regeneration actually leads to an increase in the overall flow or force generated by the cilia. ***But as the authors explain, this is probably a difficult experiment to carry out and appears to be optional in the context of this study.***

RESPONSE: We thank the reviewer for recognizing and stating that “the results are well presented and allow strong conclusions to be drawn”. We also want to sincerely thank the reviewer for understanding the technical difficulties in performing these experiments.

2. The authors are apparently only able to detect a single TZ protein, B9D1, to follow the fate of the TZ during the deciliation and reciliation process. In some ways, this provides an incomplete demonstration that all the TZ is indeed removed during deciliation, although this is supported by EM observations. It also provides a limited understanding of the time course of TZ re-formation during reciliation. Given the limitations of antibody availability, could it be possible to express tagged proteins in the animal cap system to track more TZ proteins? In particular, would it be possible to track for example Cby and NPHP proteins. What is the behavior of Cep290? This would greatly reinforce the conclusions on the molecular reorganisation of the TZ after deciliation and during cilia regeneration.

RESPONSE: We appreciate this reviewer’s brilliant questions on understanding the time course of TZ re-formation during reciliation. When we started this project and observed that TZ was lost upon deciliation in our preliminary TEM experiment, our first goal was to confirm this outstanding result. Thus, we did more TEMs and EM tomography, used bonafide TZ protein B9D1 to label the structure, and observed its loss upon deciliation. Taken together, we feel highly confident that TZ is lost upon deciliation. To address this reviewer’s concerns, we will performing additional serial TEMs to confirm the loss of TZ after deciliation.

Our next goal was to understand what the reviewer has mentioned, the TZ assembly time course. We started with TEMs at different time points and again saw a surprising result: TZ assembly was delayed compared to cilia axoneme. We were driven by this question of understanding how cilia “put together” the complex structure of TZ structurally and molecularly using EM and fluorescence data. We first attempted a few antibodies, including B9D1, CEP290, MKS5, and NPHP4, to localize to the TZ in the *Xenopus* cilia. Despite our efforts with different fixation strategies, only B9D1 appeared to localize to the TZ, whereas others did not give any signal or localized at the basal body. As per the reviewer’s request, we have tried localizing TMEM216, TMEM67, and NPHP4 using fluorescent tags, but we again found the same result: they localized to the basal body but not at the TZ. We are perplexed by this result and are pursuing the reasons behind them. In the process and additional experiments, we also realized a big pitfall in overexpressing tagged proteins during regeneration experiments. Overexpression of tagged TZ proteins may affect the native dynamics of TZ protein localization and function during regeneration. For example, endogenous B9d1 needs to be transcribed/translated post-deciliation in *Xenopus* MCCs, and overexpression of tagged B9D1 may have altered these dynamics and the conclusions. We want to note that we have used Chibby in our experiments and that it is not lost upon deciliation (Figure EV1). This is because Chibby is a distal transition fiber protein (distal end of basal body) and does not extend up to the transition zone.

However, to address the concern given technical challenges, **we have added 8 new tomograms as extended videos to support our data (Movies EV2-14)**. Some of these tomograms clearly show the complete absence of ciliary axoneme, including TZ at some basal bodies at 20min and 1 hr. Other tomograms demonstrate the TZ assembly at various stages during regeneration. We have also provided 2 TEM sections at pre-deciliation and post-deciliation (0 min. timepoint) in addition to the tomograms to show the same result (Fig. EV2E-2F).

Minor comments

3. Figure 4: The images are poorly defined, and it is difficult to distinguish individual basal bodies and cilia. Therefore, it is not clear how the authors can confidently quantify the number of basal bodies in each condition to construct the graph at the bottom of the figure. In addition, it would be interesting to label the basal body with a centriolar marker to better define it. - Figure 4 labels the Transition Zone protein B9D1 and cilia marker acetylated tubulin and not basal bodies. The graph represents the number of multiciliated cells (not basal bodies) with the presence or absence of elongated B9d1 signal.
4. Figure 5: not clear why the graph on the lower right does not include the control at 3 and 6 hrs? Is it because the number is too high and difficult to quantify? – Yes, the reviewer is right. Cilia become too long and too many to quantify their number reliably. We have mentioned this in lines 333-334 explicitly to the readers.
5. References: I would like to draw the authors' attention to studies of deciliation in Paramecia that could be cited in the introduction or discussion of the conservation of this pathway through evolution. – We have added multiple references to paramecia throughout the manuscript. Specifically, we mention that deciliation and regeneration in unicellular models like paramecia have added to our understanding of ciliogenesis. Line 106 “While it is important to remember that regeneration of cilia may not be identical to *de novo* assembly, cilia regeneration studies in *Chlamydomonas reinhardtii*, *Paramecium* and *Tetrahymena* etc., have provided significant insights into ciliogenesis, e.g., cargo transport, the presence of precursor pool, regulation of ciliary gene expression.^{18,23–26}”. Further, we also added the reference to paramecia in results, line 164 “Next, we determined the location where the deciliation treatment severed cilia. Unicellular models such as *Chlamydomonas*, *Paramecium* and *Tetrahymena* lose cilia distal to the TZ and below the central pair (CP) microtubules³³.”. We also add discussion on the importance of TZ in paramecia, line 221 “Interestingly in *Paramecium* also a unicellular multiciliated cell, displays constant shedding of cilia when TZ proteins are depleted”. These statements have been supported by the following studies that are now cited in the manuscript: Machemer and Ogura 1979 Journal of Cell Physiology (10.1113/jphysiol.1979.sp012990) and Gogenddeau et al., Plos Biology (10.1371/journal.pbio.3000640).

RESPONSE: All the minor points in the manuscript are addressed.

Reviewer 3

We thank Reviewer 3 for their time and comments. They did not have any major or minor comments.

Dear Saurabh,

Thank you for submitting your revised manuscript. It has now been seen by two of the original referees. I apologize for this unusual delay in getting back to you, it took longer than anticipated to receive the referee reports due to this busy time of the year and further discussions with the referees, as mentioned before.

As you can see, both referees find that the study is significantly improved during revision and recommend publication. However, they have outstanding concerns. In particular, referee #1 finds that additional analysis on the CHX experiments is necessary to investigate whether acetylated tubulins and IFT proteins are also expressed during regeneration, similar to B9D1. Moreover, referee #2 finds that conclusions regarding the TZ formation need to be toned down given the provided new data with other TZ components. These concerns need to be addressed for publication here. Lastly, I have further discussed the concerns of referee #2 on the quality of Figure 4, who finds the resolution of the micrographs per se sufficiently high. Referee #2 finds the quality of the micrographs per se sufficient. However, I would like to ask you to increase the quality of the labels of the images, which look somewhat pixelated. Please let me know if you need to discuss these points further.

Moreover, I need you to address the points below before I can accept the manuscript.

- Please address the remaining minor concerns of the referees and provide a point-by-point response.
- Please reduce the number of keywords to 5.
- Please replace the sentence provided in the Data Availability section with the following sentence: "This study includes no data deposited in external repositories."
- Please add a section entitled "Disclosure Statement & Competing Interests" (please see <https://www.embopress.org/page/journal/14693178/authorguide#conflictsofinterest>)
- Please provide the Author Checklist in excel format. Also, we note that the responses to the drop-down menus (of the column D) are incomplete.
- We note the following regarding the Funding information, which should be complete both in the manuscript tracking system and the manuscript text: the following is acknowledged in the manuscript, but it is missing in the manuscript tracking system: the Advanced Microscopy Facility supported by the University of Virginia School of Medicine, Research Resource Identifiers (RRID): SCR_018736) for SEM services (NIH SIG grant 1S10OD011966-01A1); NIH grant G20-RR31199. The following NIH grant is entered in the manuscript tracking system, but it is missing in the manuscript text: 1R01GM144668-01
- We note that the panels of Fig. 4 (A and B) are currently not called out in the manuscript text individually. Moreover, a callout for Fig. 8 is missing.
- Please resubmit EV Table 1 as Dataset EV1, whose legend needs to be removed from the manuscript file and provided in the table file itself. Source file name, title in the manuscript tracking system and the manuscript callouts need to be corrected accordingly.
- We note that there are two Appendix figure submitted separately. However, Appendix needs to be single PDF file. The file should have page numbers in the Table of Contents on the title page and the nomenclature in the file and manuscript callouts need to be updated to Appendix Figure S1, Appendix Figure S2, Appendix Table S1 etc. The legends of the Appendix figures should be removed from the manuscript text and included in the Appendix file.
- As for the movies, their legends need to be removed from the manuscript and each should be provided in a separate readme.txt file that should then be zipped up with the corresponding movie file and then uploaded as folder per movie (Movie EV1, etc.)
- The in-text citations for Legal et al, 2023 should be changed to preprint: Legal et al, 2023 (please see <https://www.embopress.org/page/journal/14693178/authorguide#referencesformat> for further information).
- Reagents & Tools table needs to be removed from the manuscript and uploaded separately.
- Please rename the Summary section as Abstract.
- Please rename the Materials and Methods section as Methods.
- The manuscript sections should be in the following order: Title page - Abstract & Keywords - Introduction - Results - Discussion - Methods - Data Availability - Acknowledgments - Disclosure Statement & Competing Interests - References - Figure Legends - (Main Tables with legends if applicable) - Expanded View Figure Legends.
- Our production/data editors have asked you to clarify several points in the figure legends:
 - o Please note that the exact p values are not provided in the legends of figures 1C, 2C, 3B, 4B, 5E, 7E, EV1 B, EV3 B, EV4 B.
 - o Please note that scale bar and its definition are missing for figures 2B, EV2 D
 - o Please note that the white arrows are not defined in the legend of EV1A, EV2 C. This needs to be rectified.
- Papers published in EMBO Reports include a 'synopsis' and 'bullet points' to further enhance discoverability. Both are displayed on the html version of the paper and are freely accessible to all readers. The synopsis includes a short standfirst summarizing the study in 1 or 2 sentences (max 35 words) that summarize the paper and are provided by the authors and streamlined by the handling editor. I would therefore ask you to include your synopsis blurb and 3-5 bullet points listing the key experimental findings.
- In addition, please provide an image for the synopsis. This image should provide a rapid overview of the question addressed in the study but still needs to be kept fairly modest since the image size cannot exceed 550 (width) x 300-600 (height) pixels.

Thank you again for giving us to consider your manuscript for EMBO Reports, I look forward to your minor revision.

Kind regards,

Deniz

--

Deniz Senyilmaz Tiebe, PhD
Senior Scientific Editor
EMBO Reports

Referee #1:

The authors of this manuscript addressed points from this reviewer in the revision. They provided tomograms of the base part of cilia at pre-deciliation, at deciliation and after deciliation, to highlight loss of the transition zone (TZ), recovery of the axoneme (AX) and TZ, using the *Xenopus* mucociliary epithelium. The monumental discovery is AX extension before TZ formation, which this reviewer questioned at the first stage of review, and the authors demonstrated in a convincing way by the tomograms. The extended part of the cilia, corresponding to the distal part from TZ (or H-shaped object, typical for TZ), has the central pair, indicating this part is the axoneme of motile cilia and thus AX was generated before TZ formation, as the authors claim. They also investigated the source of proteins to form TZ. They focused on B9D1. They interrupted vehicle and protein expression to follow its influence on B9D1 incorporation to cilia. They concluded that B9D1 supply depends on newly expressed proteins, not in the cytoplasmic pool to provide ciliary components. This reviewer finds their argument well demonstrated. Meanwhile it is not clear whether other ciliary components were provided from the precursor pool or require further expression. The authors should be able to interpret CHX experiments quantitatively to examine whether acetylated tubulins and IFT proteins are provided enough from the precursor pool or also expressed during regeneration. In the latter case, how much will be the ratio between proteins from the pool and newly expressed? Based on the influence of vehicle, expression and degradation interruption on the number and the length of cilia on the *Xenopus* epithelial cell, they developed discussion how cilia formation is optimized to generate fluid flow in the optimum way. The whole study is of high novelty and would deserve publication in the EMBO Reports after fulfilling the above mentioned demands and a few minor points below.

Minor comments:

Line298: "Vehicle" should be defined properly, for non-expert of cilia research.

Line345-348: Discussion here seems based on turn-over of cilia (resorption of cilia) takes place regularly at the cell. Is there any past work to prove this?

Line426-427: This sentence is not clear. Does IFT proteins decrease upon CHX treatment and then gradually increase (enrich) again? If so, this seems contradictory to the hypothesis that IFT proteins are provided from the cytoplasmic pool.

Line 447-458: This paragraph (especially the last sentence) is confusing. Their study on cilia re-generation after deciliation provides common insight into cilia assembly. Their finding that TZ is removed together with AX is specific to *Xenopus* epithelial cells, not necessarily conserved mechanism.

Line109-110: the precursor pool is very important concept in this work. However, the references are cited only broadly in this manuscript (together with other topics). Could you cite specifically the essential literature which proved the existence of precursor pool with brief explanation?

Referee #2:

The authors have partially addressed my concerns. I agree with their assessment that the EM tomograms show deciliation occurring just above the basal body (BB) and proximal to where the transition zone (TZ) should be. Regarding my suggestion to use antibodies targeting other TZ proteins, I think their observation that all other antibodies they tested for TZ components in *Xenopus* localize to the BB rather than the B9D1 compartment is significant. I believe the authors should either comment on this or soften their conclusions, stating that B9D1 is not immediately recovered, and that what they define as the TZ (the H-shaped structure) is also not immediately rebuilt. Additionally, it is important to highlight that in organisms like *Drosophila*, proteins such as B9D1, D2, and tectonic are dispensable for cilium assembly, unlike other TZ proteins like Chibby. This suggests that the MKS complex may play different roles in cilium assembly in different organisms.

I still feel that the resolution in Figure 4 does not allow us to draw conclusions with the level of precision shown in the accompanying diagrams.

I have no other concerns.

Minor points:

Apologies for not addressing this earlier, but for Figure 3 and EV3: the diagrams are somewhat misleading regarding the quantification of Senthan and Clamp length. At 1 hour, the quantification shows that neither Senthan nor Clamp length changes significantly, and then increases progressively, which doesn't match the magenta representations in the diagrams.

For Movies EV8 and EV12, it appears that the cilia are fully elongated, contrary to what is described in the legend. Could there be an error in the movie numbering?

On page 8, line 399, the sentence starting with "Moreover, in the CHX treatment..." seems incomplete.

AUTHOR RESPONSES TO REFEREE COMMENTS

We want to thank the reviewers for their service. Throughout the revision process, they identified numerous strengths, including “The monumental discovery is AX extension before TZ formation,” “The whole study is of high novelty,” and “the identification of a really cool system that promises to bring important insight to the cilia field.” In the second round of review, reviewers raised a few minor concerns. We have responded to their comments below.

Referee #1:

The authors of this manuscript addressed points from this reviewer in the revision. They provided tomograms of the base part of cilia at pre-deciliation, at deciliation and after deciliation, to highlight loss of the transition zone (TZ), recovery of the axoneme (AX) and TZ, using the *Xenopus* mucociliary epithelium. The monumental discovery is AX extension before TZ formation, which this reviewer questioned at the first stage of review, and the authors demonstrated in a convincing way by the tomograms. The extended part of the cilia, corresponding to the distal part from TZ (or H-shaped object, typical for TZ), has the central pair, indicating this part is the axoneme of motile cilia and thus AX was generated before TZ formation, as the authors claim.

They also investigated the source of proteins to form TZ. They focused on B9D1. They interrupted vehicle and protein expression to follow its influence on B9D1 incorporation to cilia. They concluded that B9D1 supply depends on newly expressed proteins, not in the cytoplasmic pool to provide ciliary components. This reviewer finds their argument well demonstrated.

We are delighted to see the reviewer finds the revisions convincing and finds our discovery monumental.

Meanwhile it is not clear whether other ciliary components were provided from the precursor pool or require further expression. The authors should be able to interpret CHX experiments quantitatively to examine whether acetylated tubulins and IFT proteins are provided enough from the precursor pool or also expressed during regeneration. In the latter case, how much will be the ratio between proteins from the pool and newly expressed?

We thank the reviewer for raising an interesting question. However, we cannot address the reviewer's question from the existing data because, unlike B9D1 and IFT, proteins accumulate at the base of cilia and are present even after deciliation. Thus, any new IFT protein cannot be distinguished from the previous protein pool without extensive experiments to label different pools with different fluorophores or by using photoconvertible fluorophores. Tubulin is the most abundant protein in the cell, and acetylation is a post-translational modification on α -tubulin. Acetylation levels can change dynamically depending on the cell's physiology/stress, etc. and do not reflect the tubulin content of the cell or cilia.

Based on the influence of vehicle, expression and degradation interruption on the number and the length of cilia on the *Xenopus* epithelial cell, they developed discussion how cilia formation is optimized to generate fluid flow in the optimum way. The whole study is of high novelty and would deserve publication in the EMBO Reports after fulfilling the above-mentioned demands and a few minor points below.

We thank the reviewer for recommending our work for publication in EMBO Reports.

Minor comments:

Line 298: "Vehicle" should be defined properly for non-experts in cilia research.

We have now indicated that DMSO serves as the vehicle in several instances, including the legends, Figures 4 and 5, and in the Results section, line 311 (previously line 298), where it is first mentioned. We have clarified this in the methods section (lines 597-599).

Line 345-348: Discussion here seems based on turn-over of cilia (resorption of cilia) takes place regularly at the cell. Is there any past work to prove this?

Now, lines 363-365. We are unaware of any specific study on cilia resorption in *Xenopus* MCCs. However, cilia resorption and disassembly due to ubiquitination are reported in *Chlamydomonas*. We have cited that specific study to support this statement.

Line 426-427: This sentence is not clear. Does IFT proteins decrease upon CHX treatment and then gradually increase (enrich) again? If so, this seems contradictory to the hypothesis that IFT proteins are provided from the cytoplasmic pool.

Thank you for this comment. IFT does not decrease and then increase in the CHX treatment. (now line 442) Once the cilia are lost, the MCCs have only the cytosolic pool available to assemble new cilia until new synthesis begins. When we block that (new synthesis by CHX), the existing pool, which was more uniformly distributed to the basal bodies, becomes enriched at a few basal bodies that have ciliary axonemes.

Line 447-458: This paragraph (especially the last sentence) is confusing. Their study on cilia re-generation after deciliation provides common insight into cilia assembly. Their finding that TZ is removed together with AX is specific to *Xenopus* epithelial cells, not necessarily conserved mechanism.

(Now line 474-476) The partial loss of TZ in human airway MCCs suggests that the mechanism of deciliation may be conserved between vertebrate MCCs. However, more studies are needed to confirm this hypothesis.

Line 109-110: the precursor pool is very important concept in this work. However, the references are cited only broadly in this manuscript (together with other topics). Could you cite specifically the essential literature which proved the existence of precursor pool with brief explanation?

The referee made a significant point, and we have now described the ciliary pool and cited the original studies in the introduction section (lines 114-125).

Referee #2:

The authors have partially addressed my concerns. I agree with their assessment that the EM tomograms show deciliation occurring just above the basal body (BB) and proximal to where the transition zone (TZ) should be.

We are glad that we could convince the reviewer that cilia are lost with TZ with the additional tomograms.

Regarding my suggestion to use antibodies targeting other TZ proteins, I think their observation that all other antibodies they tested for TZ components in *Xenopus* localize to the BB rather than the B9D1 compartment is significant. I believe the authors should either comment on this or soften their conclusions, stating that B9D1 is not immediately recovered, and that what they define as the TZ (the H-shaped structure) is also not immediately rebuilt. Additionally, it is important to highlight that in organisms like *Drosophila*, proteins such as B9D1, D2, and tectonic are dispensable for cilium assembly, unlike other TZ proteins like Chibby. This suggests that the MKS complex may play different roles in cilium assembly in different organisms.

We have now commented on the TZ protein localization at the basal bodies in our result section (line 254 - 264).

I still feel that the resolution in Figure 4 does not allow us to draw conclusions with the level of precision shown in the accompanying diagrams.
I have no other concerns.

Samples in Fig. 4 were fixed using methanol since the B9D1 antibody signal cannot be detected with other fixation methods. Methanol fixation leads to dehydration, and even after rehydration, the ciliary tuft on MCC does not appear straight as it would with PFA fixation. This does not apply to the single cilium, where we can stain using the same fixation and antibody (Fig. EV2). Unfortunately, this is a technical challenge that we cannot resolve. However, we have quantified the B9D1 length to accompany the images.

Minor points:

Apologies for not addressing this earlier, but for Figure 3 and EV3: the diagrams are somewhat misleading regarding the quantification of Senthan and Clamp length. At 1 hour, the quantification shows that neither Senthan nor Clamp length changes significantly, and then increases progressively, which doesn't match the magenta representations in the diagrams.

We have changed the magenta representations to match the quantification.

For Movies EV8 and EV12, it appears that the cilia are fully elongated, contrary to what is described in the legend. Could there be an error in the movie numbering?

The ROI for those images does not extend to the entire length of cilia, and as such, we can't say that they are fully elongated.

On page 8, line 399, the sentence starting with "Moreover, in the CHX treatment..." seems incomplete.

We have now removed this sentence.

Saurabh Kulkarni
University of Virginia, USA
1540 jefferson park avenue
virginia 22903
United States

Dear Saurabh,

Thank you for submitting your revised manuscript. I have now looked at everything and all is fine. Therefore, I am very pleased to accept your manuscript for publication in EMBO Reports.

Congratulations on a nice work!

Kind regards,

Deniz

--

Deniz Senyilmaz Tiebe, PhD
Senior Scientific Editor
EMBO Reports
